# Bidirectional generation of structure and properties through a single molecular foundation model

Jinho Chang ⓘ[1] & Jong Chul Ye ⓘ[1] ✉

Recent successes of foundation models in artificial intelligence have prompted the emergence of large-scale chemical pre-trained models. Despite the growing interest in large molecular pre-trained models that provide informative representations for downstream tasks, attempts for multimodal pre-training approaches on the molecule domain were limited. To address this, here we present a multimodal molecular pre-trained model that incorporates the modalities of structure and biochemical properties, drawing inspiration from recent advances in multimodal learning techniques. Our proposed model pipeline of data handling and training objectives aligns the structure/property features in a common embedding space, which enables the model to regard bidirectional information between the molecules' structure and properties. These contributions emerge synergistic knowledge, allowing us to tackle both multimodal and unimodal downstream tasks through a single model. Through extensive experiments, we demonstrate that our model has the capabilities to solve various meaningful chemical challenges, including conditional molecule generation, property prediction, molecule classification, and reaction prediction.

Capturing complex relations between chemical entities and their properties is the essence of numerous chemical challenges. During the last decade, artificial intelligence has emerged as a promising tool in chemistry research for estimating many biochemical properties and interactions between molecules, polymers, and proteins, which are difficult to obtain experimentally[1–3]. Various deep learning-based approaches in the chemical domain employed deep neural networks to extract desired characteristics like intrinsic properties, biochemical activities, and chemical reactions from raw molecule data[4–6]. Additionally, de novo molecule design has been extensively studied using recurrent networks[7], variational autoencoders[8,9], graph networks[10], etc[11–13]. More recently, unsupervised learning approaches of learning better representations of the chemical inputs have been suggested[14–16] to overcome the limitation of learning separate features for each task in a super- vised manner. These recent approaches are on the same track as the concept of the foundation models that are trained with large datasets and are often considered as a new paradigm of deep learning[17,18].

Specifically, a concept of pre-training a neural network in a self-supervised manner for a better feature representation has been adapted for various chemical fields[14–16]. N-Gram Graph[19] and GROVER[20] used a graph neural network and a graph transformer network, respectively, to obtain a pre-trained model from the molecular graph. ChemBERTa-2[21] trained a roBERTa model with 77 million molecules to build a molecular foundation model, by training the model to predict 200 different chemical property values.

Meanwhile, in the computer vision field, multimodal pre-training methods like Vision-Language Pre-training (VLP)[22] have achieved outstanding performance in downstream tasks that require an understanding of both image and text. Most of the modern VLP models utilize Transformer[23] architecture and its cross-attention mechanism to learn the correlation between different modalities[24,25]. Moreover, several works introduced contrastive learning, which assimilates features with the same context and distances semantically unrelated features, to align image and language features in the common feature

[1]Graduate School of AI, KAIST, Daejeon, South Korea. ✉e-mail: jong.ye@kaist.ac.kr

space[26–28]. VLP enables various tasks such as visual question answering[29], image-text retrieval[30], text-driven image generation[31], image-driven text generation[32], etc., which are not possible using single modality foundation models.

Inspired by the success of multimodal learning, several recent works tried to obtain a better feature of a molecule by leveraging knowledge from different data representations. Winter et al. trained a translation model between Simplified Molecular-Input Line-Entry System (SMILES) and International Chemical Identifier (InChI) key to get a feature vector with meaningful information that both molecular representations have in common[33]. Zhu et al. used a self-supervised training method of BYOL[34] between different molecule representations of SMILES and molecular graphs to build a dual-view model[35]. However, these works introduced multimodality only for the enhancement of a molecule feature for unimodal tasks, not for the interplay between those different modalities. Furthermore, since SMILES, InChI, and graph representations contain almost identical information about the connection between atoms in a molecule, it is unlikely to expect new emergence properties by multimodal learning between these different molecule representations.

In this work, we are interested in the cross-modal comprehension between molecule structure and the associate properties, which facilitates solving meaningful tasks in many applications like property predictions, conditional molecule design[36,37], etc. Taking a step further from multi-task learning methods[38] which use the prepared properties as labels to extract general features[21], our approach regards a set of properties as a stand-alone modality that represents the input molecule and suggests that multimodal learning for molecules with this property modality can provide much more informative features.

Specifically, we propose a molecule Structure-Property Multi-Modal foundation model (SPMM) that allows various chemistry experiments in silico, which is pre-trained with a wide range of molecules' structures and a vector of its properties. By employing a Transformer architecture[23], the intramodal feature extraction and intermodal fusion can be done with self-attention and cross-attention mechanisms, respectively.

Our experimental results show that simultaneous learning of structural features with information from the associate properties through a single foundation model gives us a better representation that can be fine-tuned for various downstream tasks. Specifically, by treating both structure and property symmetrically, the model can perform bidirectional generation and prediction with a single pre-trained model, which was not possible before.

Figure 1a illustrates the overall model architecture and training objectives for SPMM. The framework of SPMM extends the structure of the dual-stream VLP models[27,28,39]. Dual-stream VLP models encode the input for each modality with an unimodal encoder, then use another encoder module to perform cross-attention by using one modality feature as a query and the other modality feature as a key/value. When a training molecule is given, SPMM takes the molecule's SMILES string and its property vector (PV) as multimodal data inputs as shown in Fig. 1a. The SMILES and PV are passed through their corresponding unimodal encoders, which perform self-attention where embedded inputs become the key, query, and value. After two unimodal features are obtained, contrastive learning aligns the SMILES and PV features into the same embedding space by assimilating the features that contain the same context. This is known to improve the model performance by making cross-modal encoding easier and

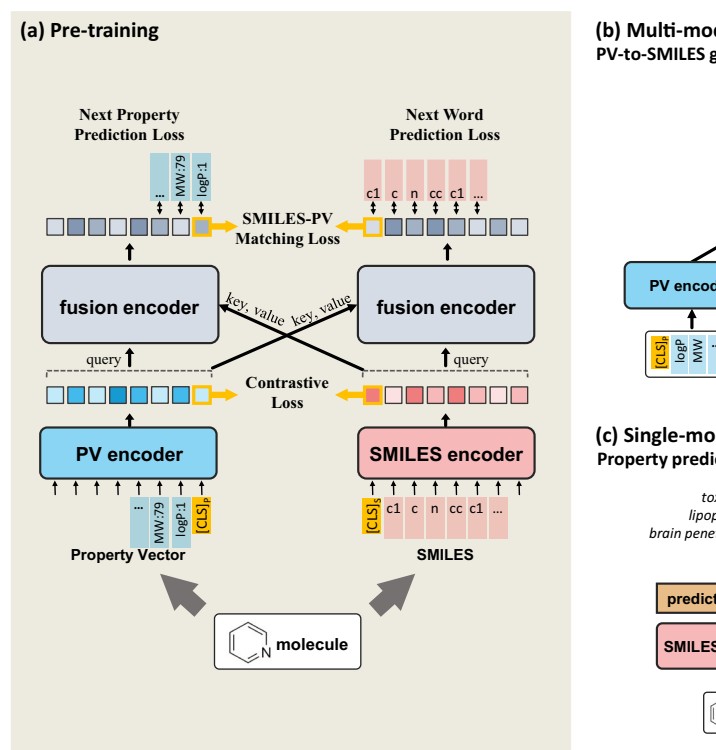

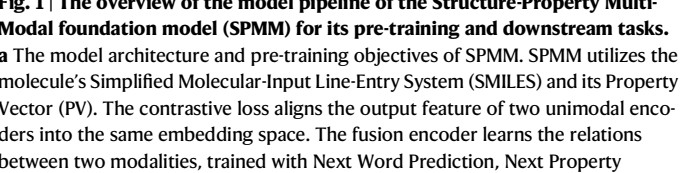

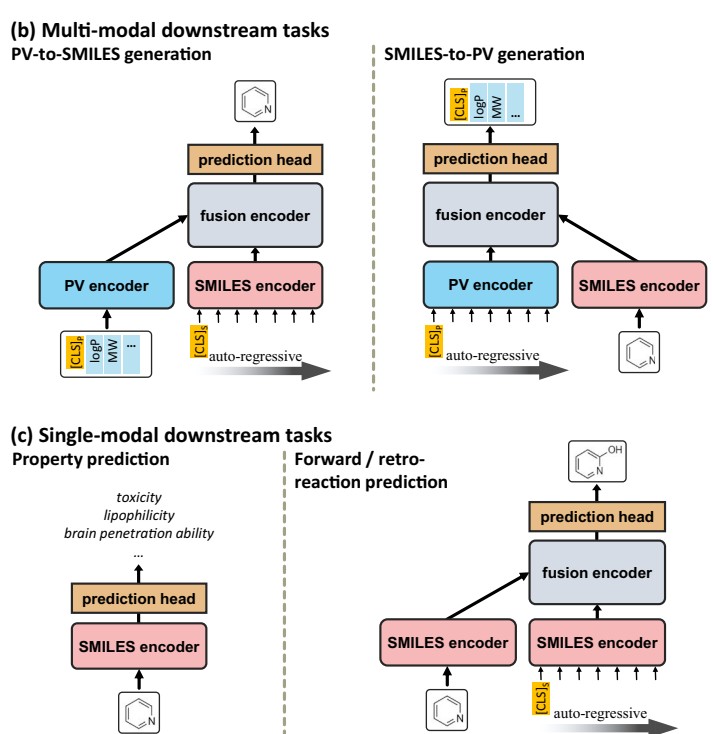

**Fig. 1 | The overview of the model pipeline of the Structure-Property Multi-Modal foundation model (SPMM) for its pre-training and downstream tasks.** **a** The model architecture and pre-training objectives of SPMM. SPMM utilizes the molecule's Simplified Molecular-Input Line-Entry System (SMILES) and its Property Vector (PV). The contrastive loss aligns the output feature of two unimodal encoders into the same embedding space. The fusion encoder learns the relations between two modalities, trained with Next Word Prediction, Next Property

Prediction, and SMILES-PV Matching loss. $[CLS]_S$ and $[CLS]_P$ represent the special token utilized for SMILES and PV modality, respectively. **b** Possible downstream task scenarios that require multimodal comprehension, namely PV-to-SMILES generation and SMILES-to-PV generation. **c** Possible downstream task scenarios for single modality inputs, namely property prediction and forward and retro reaction prediction.

guiding the unimodal encoded features to reflect more semantics of the input[27]. Then, the encoded SMILES and PV features are passed through the fusion encoders, which perform cross-attention between SMILES and PV features. This single fusion encoder can perform cross-attention with an alternation of its query and key/value input because the contrastive learning aligns the output of the SMILES encoder and the PV encoder into the same feature space[39]. The fusion encoder is pre-trained with Next Word Prediction (NWP) for SMILES, Next Property Prediction (NPP), and SMILES-PV Matching loss (SPM). Prediction of the next component from the given transformer input is a commonly used self-supervised learning objective, and our NWP and NPP tasks make the model learn the contextual relationship between SMILES tokens and properties with the aid of the other modality's semantic feature. Additionally, SPM predicts whether a given pair of SMILES and PV represents the same molecule or not.

Once trained, SPMM can be used for various bidirectional downstream tasks that require an understanding of both SMILES and properties like property prediction (SMILES-to-properties) and property-conditioned molecule generation (properties-to-SMILES, also referred to as inverse-QSAR[37]) as shown in Fig. 1b. Furthermore, the pre-training objectives that we've used allow the pre-trained SPMM to be applied for single-modality tasks as well, such as molecule classification and reaction predictions (see Fig. 1c). The pre-trained SPMM showed comparable performances to state-of-the-art models in these unimodal tasks, which suggests the model's generalization ability as a foundation model.

## Results

### The model learns bidirectional comprehension between SMILES and properties

Once SPMM was pre-trained, we made the model generate SMILES with given PV inputs only, which is a crucial challenge for many chemical tasks such as de novo molecule design. As one of the major approaches for drug discovery, various methods have been suggested for generating molecules with desired properties[9–11,13]. In the approaches presented so far, the maximum number of simultaneously controllable properties wasn't very large. Also, the length of the input property vector cannot be changed. Whenever the target properties change, the model needs to be trained again for the new wanted conditions. In contrast, the pre-trained SPMM can take 53 properties used in pre-training as input conditions and generate molecules that satisfy all of them, without separate additional training for each property combination. Moreover, for the properties that we don't want to control, we can let the model ignore those conditions by replacing them with the [UNK] token that we used in pre-training. This is very useful because controlling all 53 input properties is not a usual scenario in practice, and is also not easy since the properties are correlated and entangled (e.g., '5 atoms & 30 bonds' or '2 rings & 5 aromatic rings' is unlikely to be a valid PV input).

To demonstrate the molecule generation capability of SPMM, we prepared a number of PV-to-SMILES generation scenarios and let the pre-trained SPMM autoregressively generate SMILES using the input properties. This process of SPMM is very similar to the sequence-to-sequence translation tasks in terms of the model pipeline (see Supplementary Fig. 1a for details), from the property sentence of PV to the molecular structure sentence of SMILES.

The validity, uniqueness, and novelty of the generated molecules are the quantitative metrics of SPMM's molecule generation. Additionally, as a qualitative metric to see how the generated SMILES match the property input, we measured the normalized Root Mean Square Error (normalized RMSE) between the input conditions and the generated molecules' properties. More specifically, we calculate the average of the RMSE of all controlled properties, after those values are normalized with the corresponding property's mean and standard deviation in the pre-training dataset. We note that RMSE was calculated on the normalized scale of each property because the values of the properties span multiple orders of magnitude.

For the first PV-to-SMILES generation scenario, we prepared 1000 PVs of SMILES from PubChem[40] that are not contained in the pre-training dataset and fed them to the pre-trained SPMM to generate appropriate SMILES. Here, the sampling process was done in a deterministic manner: starting from the SMILES [CLS] token ([CLS]$_S$), the model predicts the probability distribution of the next token and chooses the option with the highest probability. The first row of Table 1 shows its results. Among the output of deterministic PV-to-SMILES generation for 1000 PVs, 99.5% of the generated output were valid SMILES. The mean RMSE of the 53 normalized properties was 0.216, which implies that the properties of the generated samples agree with the property input.

Application fields like drug discovery often require generating multiple molecules for a single wanted target property condition. This can be done by sampling the next token stochastically from the modeled probability distribution instead of using a token with the highest probability. To verify our model's ability to generate multiple molecules from a single PV input, we generated 1000 SMILES with stochastic sampling on a fixed PV. Figure 2 shows the property distributions of 1000 molecules generated from a single PV input. The mode of each property distribution lands on the input property value (Fig. 2a). In the situation when only some of the properties are given, the model only regards the known properties while the other masked properties are not restricted (Fig. 2b, c). SPMM can generate molecules even with no property information at all; when all input properties are replaced with [UNK] token (Fig. 2d), the model performs an unconditional molecule generation, and the output follows the distribution of the pre-training dataset. The validity, uniqueness, and novelty of the generated molecules under conditions in Fig. 2 are listed in the 'stochastic' rows of Table 1. The validity, uniqueness, and novelty fluctuated depending on how feasible or difficult the property input was,

**Table 1 | Quantitative and qualitative results on various scenarios of PV-to-SMILES generation tasks, with the mean value and standard deviations**

| Sampling | input PV | Validity | Uniqueness | Novelty | normalized RMSE |
|---|---|---|---|---|---|
| deterministic | 1000 unseen PubChem SMILES' PV | 0.995 ± 0.001 | 0.999 ± 0.001 | 0.961 ± 0.005 | 0.216 ± 0.004 |
| stochastic | full PV of the molecule **1** | 0.974 ± 0.005 | 0.905 ± 0.007 | 0.998 ± 0.003 | 0.185 ± 0.004 |
| | Molar mass = 150 | 0.974 ± 0.007 | 0.945 ± 0.006 | 0.872 ± 0.007 | 0.192 ± 0.010 |
| | #ring = 2, #aromatic ring = 1, TPSA = 30, QED = 0.8 | 0.998 ± 0.002 | 0.981 ± 0.006 | 0.952 ± 0.013 | 0.257 ± 0.025 |
| | no property control | 0.971 ± 0.004 | 0.991 ± 0.003 | 0.950 ± 0.003 | - |

For deterministic sampling, we ran the experiment with four different random sets of 1000 unseen Property Vector (PV)s. In the case of stochastic scenarios, four different random seeds were used for each experiment;
*TPSA* Topological Polar Surface Area, *QED* Quantitative Estimate of Drug-likeness[67], *RMSE* Root Mean Square Error.

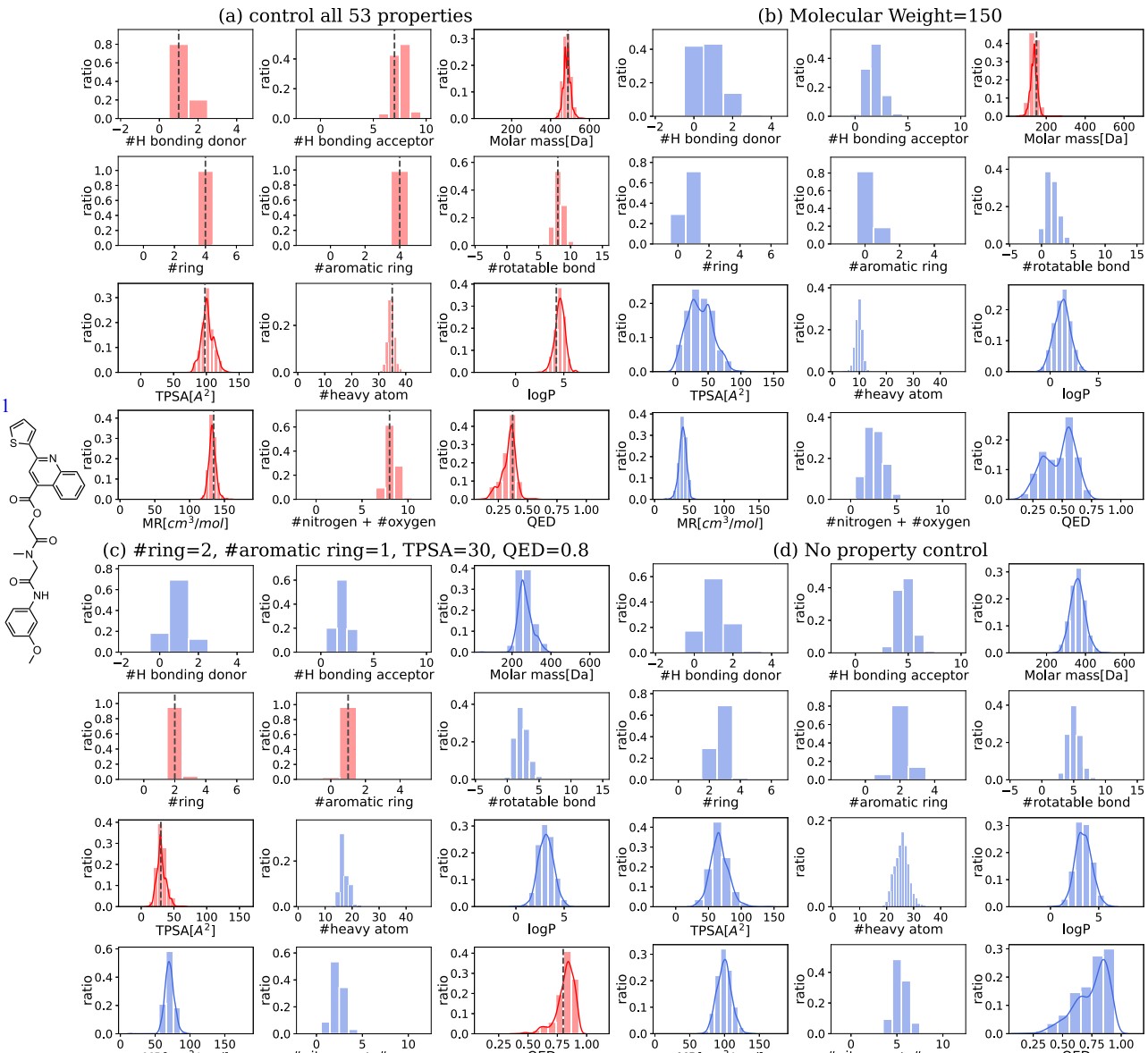

**Fig. 2 | Property distribution of the generated molecules with different Property Vectors (PV) inputs and [UNK] token masking.** The gray vertical dotted lines are the input property values. For the properties with continuous range, we included kernel density estimate plots with red or blue solid lines. The controlled properties are colored in red, and the uncontrolled properties are colored in blue. Only 12 out of 53 properties are shown for each scenario. TPSA Topological Polar Surface Area, logP octanol-water partition coefficient, MR Molar Refractivity, QED Quantitative Estimate of Drug-likeness[67]. Source data are provided as a Source Data file. **a** All 53 properties are controlled with the input PV obtained from the molecule **1**. **b** Molar mass to 150, and the other property inputs are masked with [UNK] token. **c** #ring, #aromatic ring, TPSA, and QED are controlled to 2, 1, 30, and 0.8. The other properties are masked. **d** Every property is masked.

and it was greater than 0.9 in most cases. Supplementary Table 1 shows that SPMM performed better at generating valid, novel, and desired molecules compared to other benchmark models[10,41–43], in both unconditional and conditional molecule generation scenarios. More examples of the generated molecule can be found in Supplementary Fig. 2.

The aforementioned results demonstrate that SPMM can perform molecule generation with arbitrary PV inputs, which enables simple molecule designing and editing. As possible examples of molecular editing, Fig. 3 contains the output of the SPMM's stochastic molecule generation for five PV inputs, which all originated from the PV of the molecule **1** but four of them had certain values changed. The generated molecules follow the input modification while maintaining unmodified properties similarly. SPMM is even able to generate molecules with the

out-of-domain conditions such as 'log $P = 7$' (note that ~5% of the pre-training dataset has log $P > 7$).

Regarding the overall molecule generation performance of SPMM, we want to emphasize that SPMM can generate suitable SMILES for many property conditions that the model has not seen in its pre-training. When we trained SPMM without 50% of random property masking with [UNK] token, the model only worked when all 53 properties are given since the model has not seen the partially-given properties. However, even with the technique of [UNK] token masking, the model cannot face most of the $2^{53}$ possible property combination during the pre-training process. The SPMM's ability to handle arbitrary property conditions for SMILES generation comes from treating PV as a 'language with 53 words' and focusing on each property separately, not simply considering the entire property input as a single condition.

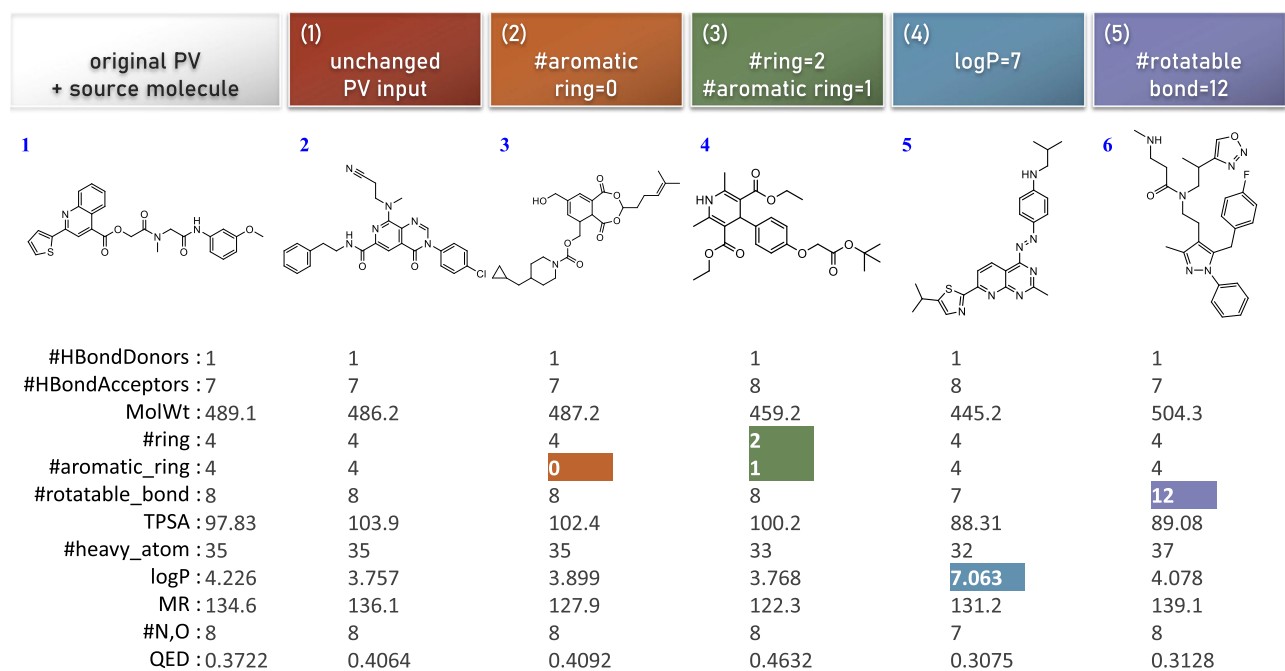

| | original PV + source molecule | (1) unchanged PV input | (2) #aromatic ring=0 | (3) #ring=2 #aromatic ring=1 | (4) logP=7 | (5) #rotatable bond=12 |
|---|---|---|---|---|---|---|
| | 1 | 2 | 3 | 4 | 5 | 6 |
| #HBondDonors : | 1 | 1 | 1 | 1 | 1 | 1 |
| #HBondAcceptors : | 7 | 7 | 7 | 8 | 8 | 7 |
| MolWt : | 489.1 | 486.2 | 487.2 | 459.2 | 445.2 | 504.3 |
| #ring : | 4 | 4 | 4 | 2 | 4 | 4 |
| #aromatic_ring : | 4 | 4 | 0 | 1 | 4 | 4 |
| #rotatable_bond : | 8 | 8 | 8 | 8 | 7 | 12 |
| TPSA : | 97.83 | 103.9 | 102.4 | 100.2 | 88.31 | 89.08 |
| #heavy_atom : | 35 | 35 | 35 | 33 | 32 | 37 |
| logP : | 4.226 | 3.757 | 3.899 | 3.768 | 7.063 | 4.078 |
| MR : | 134.6 | 136.1 | 127.9 | 122.3 | 131.2 | 139.1 |
| #N,O : | 8 | 8 | 8 | 8 | 7 | 8 |
| QED : | 0.3722 | 0.4064 | 0.4092 | 0.4632 | 0.3075 | 0.3128 |

**Fig. 3 | Examples of molecule editing, by changing specific values from the original Property Vectors (PV) and performing PV-to-SMILES generation with it.** The colored output values correspond to the changed properties from the original PV. TPSA Topological Polar Surface Area, logP octanol-water partition coefficient, QED Quantitative Estimate of Drug-likeness[67]. (1) The output of the same PV of the source molecule. (2) The output when #aromatic ring is changed to 0. (3) The output when #ring is changed to 2 and #aromatic ring is changed to 1. (4) The output when logP is changed to 7. (5) The output when #rotatable bond is changed to 12. For the generation, the other 41 property conditions are masked by the [UNK] token.

This innovative approach for conditional molecule generation has never been demonstrated with the existing methods and thus can be used for many important chemical fields.

With the same approach as SMILES generation, the pre-trained SPMM can also be used to generate a PV with SMILES input only. This task is equivalent to performing 53 property predictions of a given SMILES at once. Similar to the PV-to-SMILES generation, properties are predicted in an autoregressive manner: the model predicts the first property value using only the property [CLS] token ($[CLS]_P$), then takes all previous outputs again to get the next prediction value, and so on (see Supplementary Fig. 1b). Although 53 properties that we've used can be calculated using the Python module, the purpose of this experiment is to verify that the data-driven way of property estimation coincides with the analytic approach.

Specifically, we fed 1000 SMILES from the ZINC15 dataset[44], which are not contained in the pre-training dataset, to the pre-trained SPMM and generated their corresponding PV. Figure 4 is the scatter plot of the real property value against the generated output for 12 selected properties out of 53 that we used for pre-training. It is clear that SPMM's predicted property is very close to the actual value, and most of the data point lies on the $y = x$ line. Although the model virtually has never seen a full-filled PV in the pre-training due to the 50% of random property masking, the model could autoregressively predict all 53 properties as a whole. The mean $r^2$ score of the 53 properties was 0.924. The full scatter plot for all 53 properties with each $r^2$ score and raw RMSE is in Supplementary Figs. 3 and 4.

To provide an interpretation of the pre-trained SPMM's performance presented so far, we further analyzed the learned cross-modal comprehension between SMILES and property vectors by visualizing the attention scores from the pre-trained SPMM. Transformer-based models have the benefit of intuitive attention visualization that shows how the model considers the relation between the input queries and keys, by providing cross-attention scores between them. In Fig. 5, we plotted the cross-attention score from the last fusion layer of our pre-trained SPMM when SMILES and its property vector inputs were given.

Since there are multiple heads for the cross-attention, we took the mean of their attention scores. It is interesting that the aspect of cross-attention scores followed the intuitive relations between chemical properties and molecular fragments. The properties related to hydrogen bonding ('NumHDonors', 'NumHAcceptors') show high attention scores for tokens with oxygen and nitrogen atoms. The property 'RingCount' focuses on the tokens that are involved with rings while showing weak attention to side groups, and the property 'NumAromaticRings' only gives high attention score to the components of aromatic rings. When different SMILES tokens played a similar role in the molecule such as 'c1ccccc1' and 'c1ccccc1' in the molecule **7**, their attention patterns were similar as well. This result demonstrated that SPMM could capture the relations between molecule structures and chemical properties without explicitly-given supervision between them. For more statistical analysis, we also observed which tokens show high attention scores for 12 chosen properties, using 1000 randomly sampled molecules' cross-attention map. The result showed that tokens frequently related to certain property tend to show high attention score to that property; 'TPSA' got high attention scores towards tokens with polar atoms like oxygen and halogen atoms, 'NumHAcceptors' got tokens that involve with hydrogen bonding, and 'NumAromaticRings' got the components of aromatic rings. More detailed lists of tokens for each property can be found in Supplementary Table 2.

## Generalization ability as a molecular foundation model

So far, we have demonstrated that the pre-trained SPMM can be applied to tasks that require an understanding of the relationship between SMILES and properties. However, we can also employ the pre-trained SPMM for challenges that only use SMILES data, such as molecular property prediction. One advantage of having a dual-stream VLP model structure is that the SPMM's multimodal pre-training process includes adjusting the output of one unimodal encoder to contain contextual information from the other modality, by aligning it with the other unimodal encoder's output. This implies that the SMILES

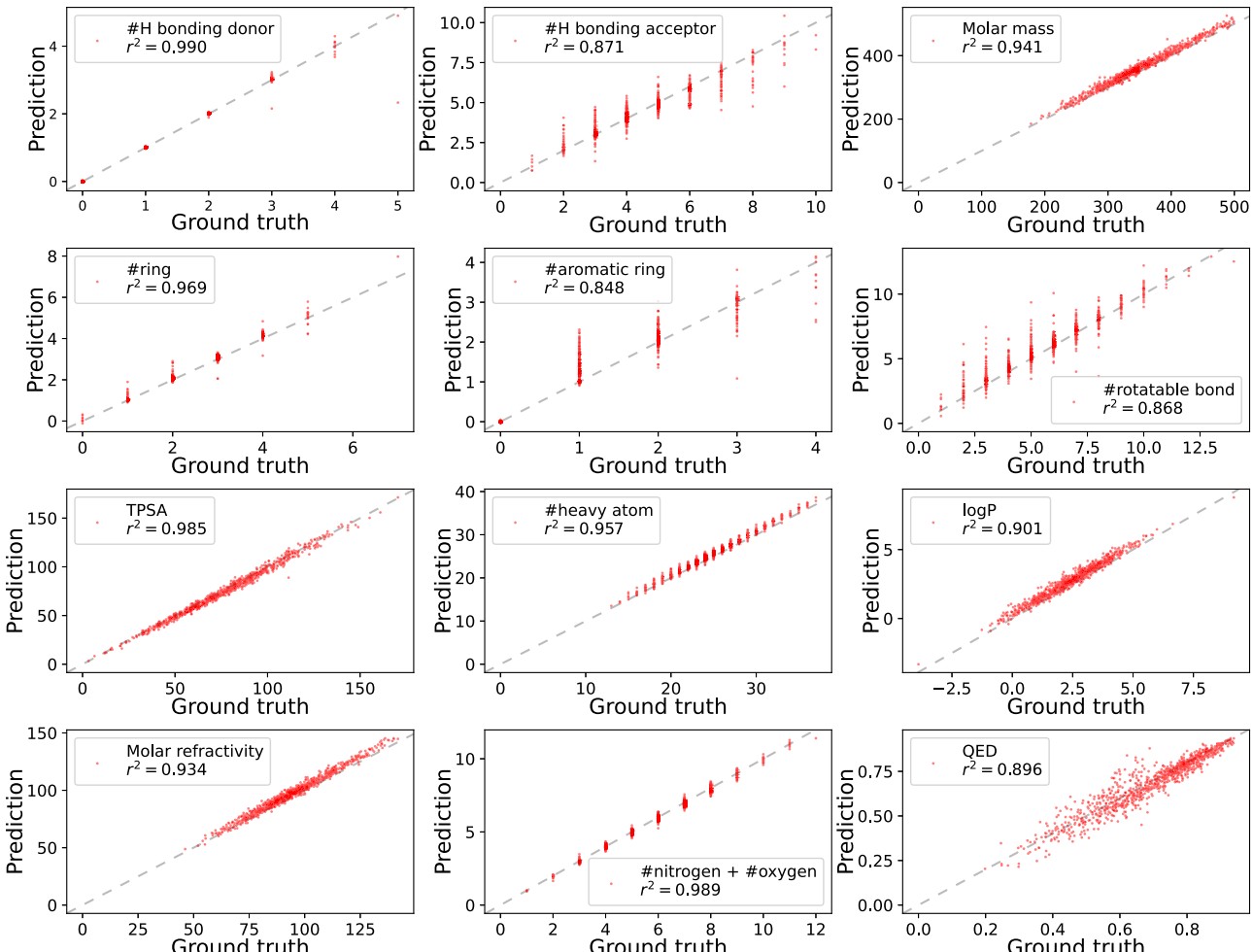

**Fig. 4 | Scatter plots of the 1000 ZINC15 molecules' real property value against the generated output, for 12 selected properties.** The *x*-axis is the real property value, and the *y*-axis is the model output. The gray dotted line is the *y* = *x* line. TPSA Topological Polar Surface Area, logP octanol-water partition coefficient. QED Quantitative Estimate of Drug-likeness[67]. Source data are provided as a Source Data file.

encoder output is a unimodal representation vector, that not only embeds the input molecule's structural information but it's also enhanced by its property information.

We have analyzed if our pre-trained model had learned an informative representation that can be readily used for other tasks, even for a single modality. So we only utilized the SMILES encoder of pre-trained SPMM (see Supplementary Fig. 1c) and made a benchmark study on nine MoleculeNet[45] downstream tasks and a Drug-Induced Liver Injury (DILI) prediction task. Each MoleculeNet task is a regression or classification task for pharmaceutical/biochemical applications like solubility, toxicity, and brain penetrability. The DILI classification task was done to overcome the potential limitation of open databases[46,47] and verify if SPMM could be extended to more complex endpoints. The task is to classify whether the given molecule has a risk of causing liver injury. Since many proposed DILI machine learning models have built their dataset rather than using common benchmarks, we took the dataset preparations from a known publication[48] and compared the performance with it for a fair evaluation.

Table 2 contains the performance of SPMM and other models for MoleculeNet. Using only 6 BERT encoder layers, SPMM showed comparable performances with state-of-the-art models for all tasks. It achieved the best performance for five tasks out of nine, showing its capability as a foundation model. We've also observed that the score of our model dramatically decreased without pre-training. SPMM also outperformed the proposed 5-ensemble models on the DILI

classification task under the same data preparation as shown in Table 3, which was not the case for the naive BERT layers without SPMM pre-training.

We also trained SPMM for the forward and retro-reaction prediction tasks, which require the model to predict the product SMILES from the reactant SMILES and vice versa. Regarding both tasks as sequence-to-sequence generation, the model pipeline for these reaction prediction tasks is the same as the PV-to-SMILES generation tasks, except the PV encoder is replaced with the SMILES encoder (see Supplementary Fig. 1d). The detailed task definition and dataset preparation are described in the Methods section.

Table 4 shows the performances of SPMM and other benchmark models on forward and retro-reaction prediction tasks. Although the reaction prediction tasks are not the best scenario for the property-emergence features to play significant roles, SPMM showed the highest top-1 accuracy in the forward-reaction task with a relatively small pre-training data size (i.e., 50 M molecules, compared to 100 M molecules of Chemformer). SPMM also achieved the second-best top-1 accuracy among the string-based retro-reaction task models.

## Discussion

In this work, we proposed a transformer-based multimodal chemical foundation model SPMM. The proposed model allows for bidirectional generation/prediction of molecular structure and properties, as well as unimodal tasks like reaction prediction. During the process, we

**(a)**

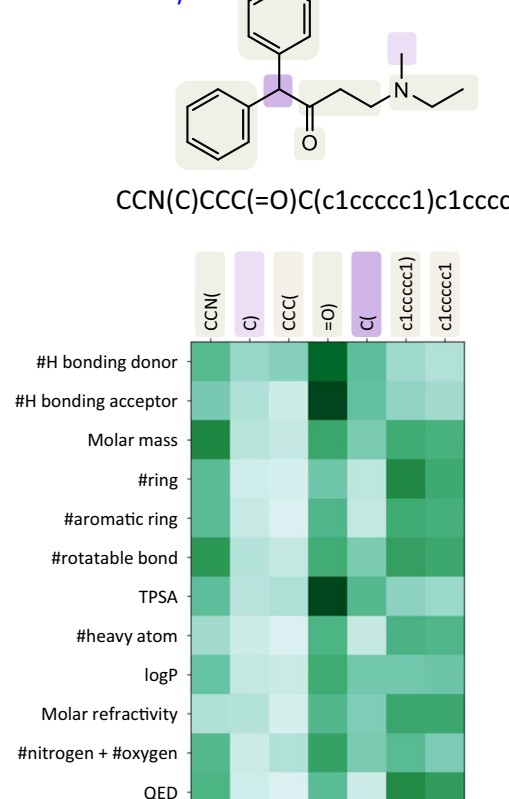

CCN(C)CCC(=O)C(c1ccccc1)c1ccccc1

**(b)**

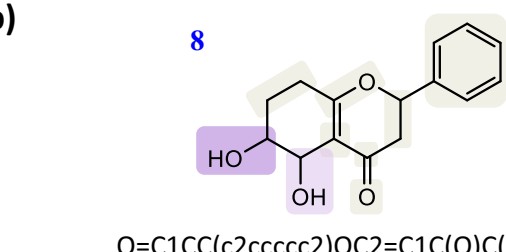
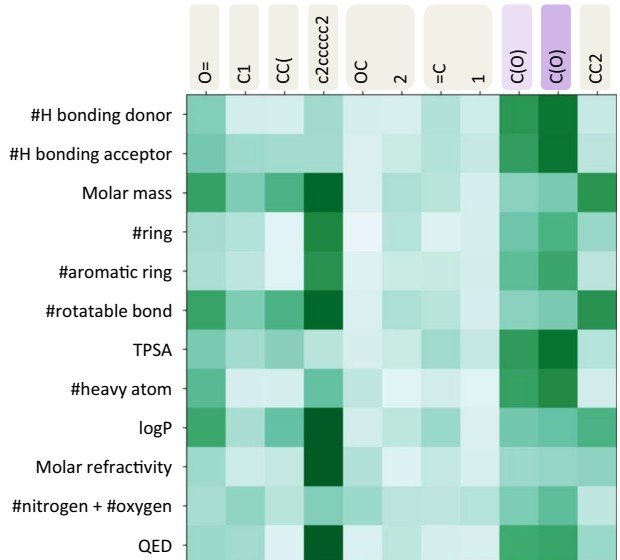

O=C1CC(c2ccccc2)OC2=C1C(O)C(O)CC2

**Fig. 5 | The mean attention score from the attention heads in the Structure-Property Multi-Modal foundation model (SPMM) fusion encoder's final cross-attention layer.** Two sample molecules (**a**) and (**b**) were used for this figure. A darker green means a higher attention score. For the attention calculation process, the property features were used as queries, and the SMILES features were used as keys and values. The corresponding fragments for each token are indicated with ivory boxes on the molecular structure, while fragments for duplicated tokens are color-coded with purple. We have calculated cross-attention scores for all 53 properties and SMILES tokens, but only 12 of those properties are shown. TPSA Topological Polar Surface Area, logP octanol-water partition coefficient, QED Quantitative Estimate of Drug-likeness[67]. Source data are provided as a Source Data file.

introduced a method of treating property collections as a language so that the model could learn the relationship between SMILES tokens and each property independently. We demonstrated that pre-trained SPMM showed remarkable performances in problems for interactions between SMILES and PV domains. And not only for multimodal challenges but even its unimodal feature for SMILES, SPMM also provides a useful representation that can be fine-tuned for many molecular downstream tasks. It is important to note that all of these results were obtained with a pre-training of 50 million molecules, which is relatively small compared to other large pre-training approaches and still has room for better performance with more data and parameters. We also note that we've gathered our 53 properties to let them cover the widest range possible, rather than paying the best effort to select the most effective combination of properties. This implies the proposed structure-property multimodal training can be flexibly adopted with different property selections, according to the given specified scenarios.

One might consider that treating a PV as tabular data, handling its elements without predetermined order, could be a more straightforward approach. However, the recent theoretical work[49] showed that autoregressive modeling is a universal learner that is not specific to any data type. Moreover, the transformer architecture's permutational invariance of the positional encoding has been well documented and utilized[50,51]. Additionally, if the order of PVs is permuted in a different order, then the learnable positional embedding would learn different embedding that takes into account the optimal alignment between the properties and the SMILES embeddings. This makes the output of the transformer's self-attention and cross-attention mechanism is further invariant to the position of each feature vector. Given all this evidence, we believe that our way of utilizing the PV encoding is theoretically and empirically well-supported. In fact, when we modified the training objectives (more details in the Pre-training objectives section) to pre-train our model with a purely order-invariant PV, this did not improve performance on the overall downstream tasks (see Supplementary Table 3). Moreover, we observed that utilizing different property orders for constructing PVs does not affect the overall performance of SPMM (see Supplementary Table 4).

Despite the noticeable performances of SPMM, it has several chances for improvement. One of those comes from using the SMILES notation. Although SMILES can contain full details about the 2D structure of the molecule, the information on how atoms and bonds are connected only exists implicitly. Also, a slight modification in molecular structure can be a drastic change in SMILES. Graph format is another widely used modality for molecule representation that contains the explicit information of the adjacency matrix, which can be an alternative for SMILES. Another limitation in our current SPMM is that the 53 properties we used happen to be invariant with the changes in the stereochemistry of the given molecule. It is known that considering stereochemistry plays a crucial part in various biochemical tasks. However, the 53 properties we used cannot provide any knowledge about stereochemical information since their values are unchanged in different stereoisomers. This makes the SMILES encoder output of different stereoisomers converge since the contrastive loss aligns them to the same PV feature. We believe this is the prominent factor

**Table 2 | Benchmark results on MoleculeNet downstream tasks**

| Dataset | regression[RMSE, ↓] | | | | | classification[AUROC in %, ↑] | | | |
|---|---|---|---|---|---|---|---|---|---|
| | Delaney ESOL | LIPO | Freesolv | BACE | Clearance | BBBP | BACE | Clintox | SIDER |
| #data | 1128 | 4200 | 642 | 1513 | 837 | 2039 | 1513 | 1478 | 1427 |
| #task | 1 | 1 | 1 | 1 | 1 | 1 | 1 | 2 | 27 |
| D-MPNN[68] | 1.050 ± 0.008 | 0.683 ± 0.016 | 2.082 ± 0.082 | 2.253[a] | 49.754 | 71.0 ± 0.3 | 80.9 ± 0.6 | 90.6 ± 0.6 | 57.0 ± 0.7 |
| N-GramRF[19] | 1.074 ± 0.107 | 0.812 ± 0.028 | 2.688 ± 0.085 | 1.318[a] | 52.077[a] | 69.7 ± 0.6 | 77.9 ± 1.5 | 77.5 ± 4.0 | 66.8 ± 0.7 |
| N-GramXGB[19] | 1.083 ± 0.082 | 2.072 ± 0.030 | 5.061 ± 0.744 | - | - | 69.1 ± 0.8 | 79.1 ± 1.3 | 87.5 ± 2.7 | 65.5 ± 0.7 |
| PretrainGNN[69] | 1.100 ± 0.006 | 0.739 ± 0.003 | 2.764 ± 0.002 | - | - | 68.7 ± 1.3 | 84.5 ± 0.7 | 72.6 ± 1.5 | 62.7 ± 0.8 |
| GROVER$_{large}$[20] | 0.895 ± 0.017 | 0.823 ± 0.010 | 2.272 ± 0.051 | - | - | 69.5 ± 0.1 | 81.0 ± 1.4 | 76.2 ± 3.7 | 65.4 ± 0.1 |
| ChemRL-GEM[70] | 0.798 ± 0.029 | 0.660 ± 0.008 | 1.877 ± 0.094 | - | - | 72.4 ± 0.4 | 85.6 ± 1.1 | 90.1 ± 1.3 | 67.2 ± 0.4 |
| ChemBERTa-2$_{(MTR-77M)}$[21] | 0.889[a] | 0.798[a] | - | 1.363[a] | 48.515[a] | 72.8[a] | 79.9[a] | 56.3[a] | - |
| MolFormer[71] | 0.880 ± 0.028 | 0.700 ± 0.012 | 2.342 ± 0.052 | 1.047 ± 0.029 | 43.175 ± 1.537 | 73.6 ± 0.8 | 86.3 ± 0.6 | 91.2 ± 1.4 | 65.5 ± 0.2 |
| SPMM(w/o pre-train) | 1.272 ± 0.015 | 1.009 ± 0.021 | 3.018 ± 0.179 | 1.675 ± 0.010 | 53.544 ± 0.312 | 66.6 ± 0.3 | 78.7 ± 2.6 | 76.3 ± 1.5 | 57.1 ± 1.6 |
| SPMM | 0.817 ± 0.010 | 0.681 ± 0.004 | 1.868 ± 0.004 | 1.041 ± 0.022 | 42.607 ± 0.675 | 75.1 ± 0.9 | 84.4 ± 0.4 | 92.7 ± 0.7 | 66.9 ± 0.9 |

For each task, we fine-tuned our model in four random seeds and recorded the mean and the standard deviation of those results. The benchmark model results were taken from ChemRL-GEM and ChemBERTa-2.
RMSE Root Mean Square Error, AUROC Area Under Receiver Operating Characteristic curve.
[a]The standard deviation cannot be found in the source of the benchmark results.
[b]Unofficial results, obtained from the official checkpoint under our data preparation.

that lowered the performance of SPMM in MoleculeNet tasks, which could be resolved by using more properties that reflect the molecule's stereochemistry. Moreover, validation through wet-lab experiments to verify the model's predicted/generated properties is another possible further study. Overcoming these drawbacks of the current study and making the model more applicable to other chemical tasks could be the works for the future.

Nevertheless, we believe that our approach can provide a pre-trained model capable of encompassing each input domain and their multimodal domain simultaneously, which has a vast potential utility. We expect this approach to be applied to more various and practical chemical situations by using broader and richer molecular modalities, and possibly, different biochemical domains like polymers and proteins.

## Methods

### Handling SMILES and property values as a language

Molecules can be represented with various formats such as finger-prints, strings like SMILES, InChI, or a molecular graph. Since these different notations contain almost the same information about complete molecular structure, we employed SMILES to describe a molecule structure. SMILES is a sequence of characters that represents the connection structure of the molecule. Many researchers treat SMILES as a variant of language data and utilize a concept of language models for chemical tasks on SMILES data[11,21,52].

Figure 6a illustrates our embedding procedure for the input SMILES. The raw SMILES string is tokenized by the tokenizer and embedded by the SMILES encoder with the $[CLS]_S$ token and the [SEP] token. Here, [CLS] token is a special token attached to the beginning of every input sequence[53]. Although the [CLS] token itself doesn't contain any meaning, the bidirectional attention mechanism of the model allows the [CLS] token to contain contextual information of the entire input. Once the model is pre-trained, the [CLS] token output of the given sequence can be considered as an input representation vector and be used for classification/regression downstream tasks, as in many BERT variations for images[54,55] and VLP[27].

In the SMILES tokenization, our tokenizer tokenizes a given SMILES into fragments that are contained in a prepared token dictionary of 300 subwords. This dictionary was obtained from the pre-training data SMILES corpus by the BPE algorithm[56], which starts from a set of simple characters and iteratively appends the most frequent token pairs as a merged subword. Being widely adopted for various language models[57,58], the BPE algorithm has provided a subword dictionary containing common functional groups and substructures like benzene rings, carbonyl groups, two-letter atoms, and amino groups. Compared to naive character-wise tokenization which considers each character as a separate token, the merged subwords help the model's chemical inference for chemical groups and reduce the total number of tokens.

For this work, we built a PV for each molecule that contains 53 molecular properties and considered this as a sentence with a length of 53. These properties from the RDKit python library[59] cover a wide range from simple ones, such as the number of rings and molar mass, to complex properties like solubility, TPSA, and druggability. The transformer architecture of our model considers each element of PV as a token to perform the attention mechanism, which is equivalent to regarding PV as a semi-sentence of 53 properties. Although the size of the vocabulary is more limited and their order is fixed compared to natural language, it provides much more precise and compact information about the 53 properties. One benefit of regarding PV as a language is that we do not have to collect all elements to build a valid PV. In contrast to a simple vector input, some property elements can be removed or masked in our approach.

Figure 6b shows our embedding procedure for the input PV. Each property element in the PV is a numerical value and normalized with

**Table 3 | The Drug-Induced Liver Injury (DILI) classification task performance of Ai et al.[48] and the Structure-Property Multi-Modal foundation model (SPMM)**

| model | Acc in %[↑] | Selectivity in %[↑] | Specificity in %[↑] | AUROC in %[↑] |
|---|---|---|---|---|
| Ai et al.[48] (best single model on training set) | 81.1 | 81.0 | 81.5 | 89.6 |
| Ai et al. (5-ensemble) | 84.3 | 86.9 | 75.4 | 90.4 |
| SPMM(w/o pre-train) | 72.6 | 70.6 | 79.2 | 82.0 |
| SPMM | 84.4 | 83.9 | 84.6 | 92.6 |

**Table 4 | The performance of the Structure-Property Multi-Modal foundation model (SPMM) and other works on the forward and retro-reaction prediction task**

| forward prediction | molecule modality | | top-k accuracy in %[↑] | | | |
|---|---|---|---|---|---|---|
| | string-based | graph-based | k = 1 | k = 2 | k = 3 | k = 5 |
| Molecular Transformer[72] | O | | 88.7 | 92.1 | 93.1 | 94.2 |
| Augmented Transformer[63] | O | | 90.6 | 94.4 | - | 96.1 |
| Chemformer$_{large}$[64] | O | | 91.3 | - | - | 93.7 |
| Graph2SMILES[73] | O | O | 90.3 | - | 94.0 | 94.8 |
| MEGAN[74] | | O | 86.3 | 90.3 | 92.4 | 94.0 |
| LocalTransform[6] | | O | 90.8 | 94.8 | 95.7 | 96.3 |
| SPMM | O | | 91.5 | 93.5 | 94.6 | 95.4 |

| retro-reaction prediction | top-k accuracy in %[↑] | | |
|---|---|---|---|
| | k = 1 | k = 5 | k = 10 |
| SCROP[75] | 43.7 | 65.2 | 68.7 |
| Two-way Transformer[76] | 47.1 | 73.1 | 76.3 |
| Augmented Transformer[63] | 48.3 | 73.4 | 77.4 |
| Chemformer$_{large}$[64] | 54.3 | 62.3 | 63.0 |
| SPMM | 53.4 | 67.6 | 70.3 |

For the retro-reaction prediction task, we only prepared the benchmark results of string-based models. The benchmark model results are from the paper of LocalTransform[6] and Chemformer[64].

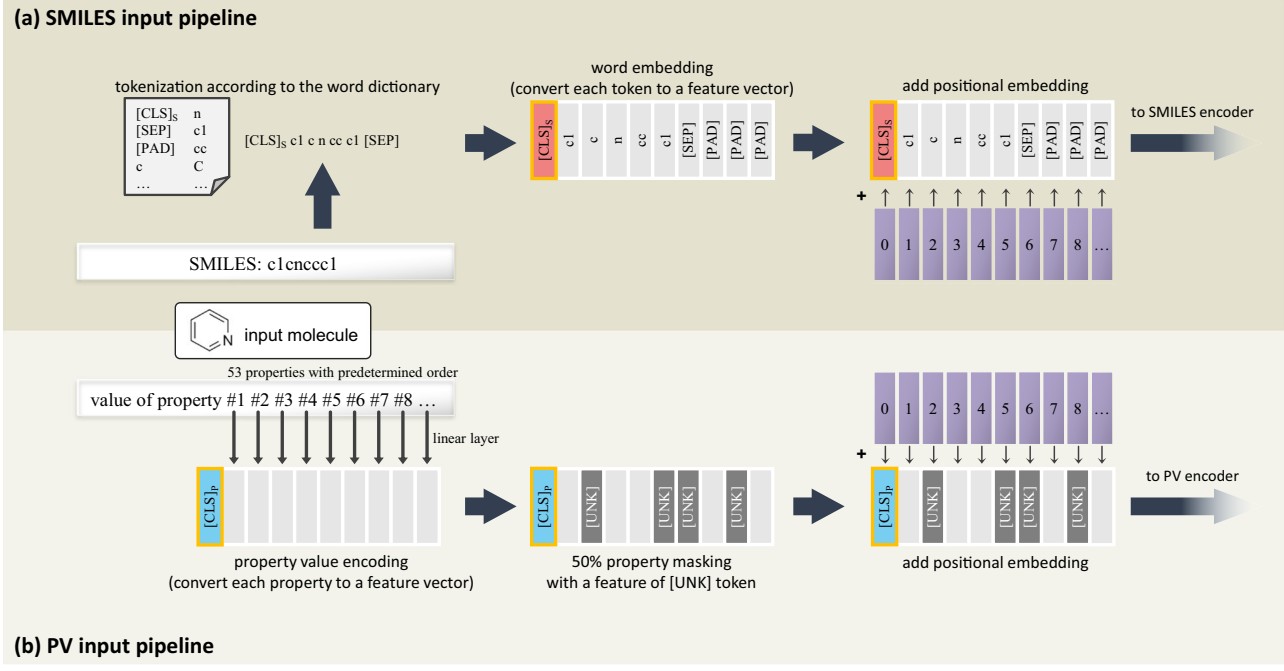

**Fig. 6 | Embedding process for SMILES and the corresponding Property Vector(PV). a** The SMILES representation of the input molecule is tokenized with the word dictionary, which was obtained with the BPE algorithm. Each token in the tokenization result is replaced with its corresponding word embedding, then passed to the SMILES encoder with positional embedding added. **b** A list of the molecule's chemical properties is prepared and passed through a linear layer. The output vectors are randomly replaced with special [UNK] token, and the result is passed to the PV encoder with positional embedding.

the mean and standard deviation of that property. The order of these 53 properties is predetermined, as Supplementary Table 4 shows that the performance of SPMM isn't affected by certain property orders used in the PV. Each value in the PV is encoded to a feature vector using a linear layer as a value encoding. Then we randomly replace 50% of the property features into the [UNK] token, which is the special token utilized to simulate that the property is unknown. This is possible since there is no problem in describing a molecule using only a part of these properties. Random property feature masking prevents the model from overly dependent on the specific property, has the effect of data augmentation, and improves the model's generalization ability. Although every property we used in this work can be easily and thoroughly prepared by the computer, this might not be the case for other properties in real-world situations. SPMM still can be trained when some properties for certain training molecules are not known, by replacing those unknown properties with the [UNK] token. On top of the randomly-masked value encoding, we added a learnable positional encoding similar to that in BERT. Since a PV explicitly contains the values only, this positional embedding provides information about what property each value corresponds to. Also, because of the predefined order of these properties, this position embedding is equivalent to giving a unique index for each property and adding an embedding of that corresponding index. Then we pass the final result to the PV encoder with the [CLS]$_P$ token.

## Pre-training objectives

Contrastive learning aims to learn better unimodal representation by aligning the features from different modalities into the same feature space[26]. When the encoded features of [CLS] tokens of SMILES **S** and PV **P** are given as $\mathbf{S}_{cls}$ and $\mathbf{P}_{cls}$, we calculate the similarity function sim(**S**, **P**) and sim(**P**, **S**) as:

$$\text{sim}(\mathbf{S},\mathbf{P}) = (h_S(\mathbf{S}_{cls}))^{\top} h_P(\mathbf{P}_{cls}), \text{sim}(\mathbf{P},\mathbf{S}) = (h_P(\mathbf{P}_{cls}))^{\top} h_S(\mathbf{S}_{cls}) \quad (1)$$

where $h_S$ and $h_P$ are the linear projection + normalization layer for SMILES and property vector, respectively. Now, for a given pair of **S** and **P**, we calculate the SMILES-to-PV and PV-to-SMILES intermodal similarities as follows[26,27]:

$$s_{s2p} = \frac{\exp(\text{sim}(\mathbf{S},\mathbf{P})/\tau)}{\sum\limits_{n=1}^{N} \exp(\text{sim}(\mathbf{S},\mathbf{P}_n)/\tau)}, s_{p2s} = \frac{\exp(\text{sim}(\mathbf{P},\mathbf{S})/\tau)}{\sum\limits_{m=1}^{M} \exp(\text{sim}(\mathbf{P},\mathbf{S}_m)/\tau)} \quad (2)$$

where $M$ and $N$ are the total numbers of SMILES and PV used in the loss calculation. Here, $\tau$ is a learnable temperature parameter, which has a sharpening effect by exaggerating the similarity difference. The intramodal similarities can be calculated in the same way.

$$s_{s2s} = \frac{\exp(\text{sim}(\mathbf{S},\mathbf{S})/\tau)}{\sum\limits_{m=1}^{M} \exp(\text{sim}(\mathbf{S},\mathbf{S}_m)/\tau)}, s_{p2p} = \frac{\exp(\text{sim}(\mathbf{P},\mathbf{P})/\tau)}{\sum\limits_{n=1}^{N} \exp(\text{sim}(\mathbf{P},\mathbf{P}_n)/\tau)} \quad (3)$$

The overall contrastive loss is defined using the cross-entropy loss $H$ and one-hot similarity $y$, which contains 1 for pairs originating from the same molecule and contains 0 otherwise.

$$L_{contrastive} = \frac{1}{2}(H(y_{s2p}, s_{s2p}) + H(y_{p2s}, s_{p2s}) + H(y_{s2s}, s_{s2s}) + H(y_{p2p}, s_{p2p})) \quad (4)$$

Following the recent contrastive loss application in VLP[60], we build the SMILES and PV queues that store the $k$ most recent SMILES and PV instances and use them for contrastive loss. We set our queue size $k$ to 24,576.

Next Word Prediction (NWP) trains the model to predict the $(n+1)$-th SMILES token when $0 \sim n$-th tokens and the corresponding PV

are given. Predicting the next token is a common objective for training language models, known for being utilized in the pre-training of GPT[61]. This can be done with a single flow for each SMILES by applying a causal mask in the self-attention of the SMILES encoder and the fusion encoder. Let $\mathbf{S} = \{s_0, s_1, ..., s_n\}$ and **P** denote the input SMILES and the corresponding PV, and $p^{NWP}(s_n|s_{0:n-1}, \mathbf{P})$ denote the model's predicted probability distribution of the $n$-th token with given **P** and 0 - $(n$-1)-th SMILES tokens. The loss for NWP is defined as follows:

$$L_{NWP} = \sum_{i=1}^{n} H(y_n^{NWP}, p^{NWP}(s_n|s_{0:n-1}, \mathbf{P})) \quad (5)$$

where $y_n^{NWP}$ is a one-hot label for the $n$-th SMILES token $s_n$.

We applied a similar concept of NWP for the property vector as Next Property Prediction (NPP). NPP makes the model predict the next property value using its corresponding SMILES and the previous properties. Since each property element is a numerical value, we replaced the cross-entropy loss in NWP with mean-square-error loss. When **S** and $\mathbf{P} = \{p_0, p_1, ..., p_n\}$ denotes the input SMILES-PV pair and $\hat{p}_n(p_{0:n-1}, \mathbf{S})$ denotes the model's predicted next property values with causal mask in the PV and the fusion encoder, the loss for NPP is given as follows:

$$L_{NPP} = \sum_{i=1}^{n} (p_n - \hat{p}_n(p_{0:n-1}, \mathbf{S}))^2 \quad (6)$$

In NPP, the model does not predict the property value if it is replaced with [UNK] token.

SMILES-PV Matching (SPM) learns if a given SMILES-PV pair (**S**, **P**) is matched or not. We concatenate the feature of [CLS]$_S$ and [CLS]$_P$ token from the fusion encoder output and pass this through a linear-layer SPM head. When $p^{SPM}(\mathbf{S}, \mathbf{P})$ is the output of the SPM head, the SPM loss can be defined as

$$L_{SPM} = H(y^{SPM}, p^{SPM}(\mathbf{S},\mathbf{P})) \quad (7)$$

where $y^{SPM}$ is a *one*-hot vector for a binary label of SMILES-PV matching; the label is 1 if **S** and **P** originated from the same molecule and 0 otherwise. To build negative samples for SPM, we randomly select a 'negative' pair for each SMILES and PV instance from the other modality and match them as negative pairs. This negative pair selection was done by hard-negative mining, which gives a higher chance of being selected as a negative pair for instances that has a higher similarity of Eq. (2) but isn't a positive match. This makes the training more difficult and forces the model to learn how to distinguish similar instances.

In contrastive learning, using a one-hot label could be too strict since it regards all instances that came from other pairs as equally negative instances. However, some PVs might agree with many SMILES, not only one SMILES that they're paired with. Even SMILES can be matched with different PVs since there's a 50% of masking in a PV (e.g., 'molar mass = [UNK], logP = 2.1, #atom = 12' and 'molar mass = 78, logP = 2.1, #atom = [UNK]' both explain Benzene, even if they came from different molecules). A similar problem also occurs for NWP. Sometimes there could be multiple sensible options for being the next token, but using a one-hot label for ground truth might ignore this.

To resolve this issue, we built the momentum teacher model[27,60] and utilized its output for contrastive learning and NWP. The momentum teacher performs a knowledge distillation by providing a pseudo-label that reflects how the teacher model comprehends. Specifically, the label for the contrastive learning and NWP are mixed with the momentum model's output $s_{*,momentum}(* \in \{s2p, p2s, s2s, p2p\})$ and $p^{NWP}_{momentum}(s_n|s_{0:n-1}, \mathbf{P})$, with an adjusting hyperparameter $\alpha$. The detailed formulas for utilizing the momentum model for contrastive

learning and NWP are described in Eq. (8)~(9) and Eq. (10)~(11).

$$\tilde{y}_* = (1-\alpha)y_* + \alpha s_{*,momentum} \ (* \in \{s2p, p2s, s2s, p2p\}) \quad (8)$$

$$\tilde{L}_{contrastive} = \frac{1}{2}\left(H\left(\tilde{y}_{s2p}, s_{s2p}\right) + H\left(\tilde{y}_{p2s}, s_{p2s}\right) + H\left(\tilde{y}_{s2s}, s_{s2s}\right) + H\left(\tilde{y}_{p2p}, s_{p2p}\right)\right) \quad (9)$$

$$\tilde{y}_n^{NWP} = (1-\alpha)y_n^{NWP} + \alpha p_{momentum}^{NWP}(s_n|s_{0:n-1}, \mathbf{P}) \quad (10)$$

$$\tilde{L}_{NWP} = \sum_{i=1}^{n} H\left(\tilde{y}_n^{NWP}, p^{NWP}(s_n|s_{0:n-1}, \mathbf{P})\right) \quad (11)$$

After the student model's parameters $w_{model}$ are updated for each batch, the parameters of the momentum teacher model $w_{momentum}$ are updated by the exponential moving average (EMA) using $w_{model}$ and an EMA hyperparameter $\lambda$ according to Eq. (12).

$$w_{momentum} = (1-\lambda)w_{model} + \lambda w_{momentum} \quad (12)$$

The overall pre-training objective is the combined loss of Contrastive, NWP, NPP, and SPM loss.

$$L = \tilde{L}_{contrastive} + \tilde{L}_{NWP} + L_{NPP} + L_{SPM} \quad (13)$$

We note that when there's no specific attention mask (e.g., causal attention mask as in GPT[61]), the output of the transformer's self-attention and cross-attention mechanism is invariant to the position of each feature vector. This means the only pre-training objective that the predetermined order or the PV's 53 properties matters is the NPP task, and SPMM behaves identically for the other pre-training objectives when we permute the order of the properties and their corresponded position embedding. If we replace the NPP task to an order-invariant objective (e.g., masked language modeling of BERT[53]), the predetermined order of the elements in a PV would not affect the output of SPMM at all. Supplementary Table 3 shows the performance of SPMM when the NPP task was replaced to masked property modeling, and we found that this doesn't improve the model performance.

## Training for downstream tasks

Supplementary Fig. 1 describes how we utilized our pre-trained model for downstream tasks. For PV generation and SMILES generation (Supplementary Fig. 1a, b), we don't need additional fine-tuning since their training objectives are already included in the pre-training (NWP, NPP). For the inference procedure, the model generates PV or SMILES with autoregressive sampling. Specifically, starting from the [CLS] token of the modality that we want to generate, the model predicts the first component and repeats taking the previous outputs to predict the next component until it's done or meets a sign to stop. For PV-to-SMILES generation, we used a beam search of $k = 2$ to help the model generate valid SMILES. The metrics of validity, uniqueness, and novelty that we've used for PV-to-SMILES generation are defined as follows:

$$validity = \frac{\#SMILES\ with\ valid\ syntax}{\#generated\ SMILES} \quad (14)$$

$$uniqueness = \frac{\#non\ duplicate\ valid\ SMILES}{\#valid\ SMILES} \quad (15)$$

$$novelty = \frac{\#unique\ SMILES\ not\ in\ the\ pretraining\ data}{\#unique\ SMILES} \quad (16)$$

For MoleculeNet downstream tasks and the DILI classification task that only provide SMILES data, we utilized only the SMILES encoder part of the model (Supplementary Fig. 1c). After the input molecule is encoded with the SMILES encoder, we pass the feature of the [CLS]$_S$ token through a classification/regression head to get an output. The classification/regression head consists of MLP with one hidden layer. We fine-tuned our model with the given training set and get a checkpoint with the lowest loss on the validation set, and recorded that checkpoint's performance on the test set.

The forward reaction prediction task provides a reactant SMILES (including multiple reagent molecules) and a product SMILES. We encode these two inputs with the SMILES encoder, then feed them into the fusion encoder + prediction head. The model is trained to auto-regressively generate the original product SMILES (Supplementary Fig. 1d). In the inference stage, starting from the [CLS]$_S$ token, the model predicts the next token until it generates the [SEP] token. Similar to the SMILES generation, the self-attention of the fusion encoder and the reactant SMILES encoder uses a causal mask. The retro-reaction prediction task was done in the same way, but the role of the reactant and product SMILES were swapped. We fine-tuned SPMM for the forward reaction prediction task with an approach of 'mixed task', meaning that the information about the major reactant is not given to the model. For both forward and retro-reaction tasks, we replaced the input reactants and products with their random non-canonical augmented SMILES[62] with a probability of 0.5. This SMILES augmentation is reported[63,64] to increase the accuracy of sequence-based reaction prediction models, and we listed the ablation study results about this in Supplementary Table 5.

## Data preparation

We obtained 50,000,000 SMILES of general molecules from PubChem[40] for pre-training. All 53 properties we used can be calculated with SMILES using the RDKit Python library[59]. The dataset for the MoleculeNet downstream tasks is provided by the DeepChem[65] Python library. We split every dataset into train/valid/test sets in a ratio of 8:1:1 using a scaffold splitter from DeepChem, which is a more harsh condition for the model than random splitting. For the reaction prediction task, we used the USPTO-480k dataset which contains 479,035 pairs of reactants and the major product of their reaction. The retro-reaction prediction task used the USPTO-50k dataset, containing 50,037 product-reactant pairs with corresponding reaction types. Although the USPTO-50k dataset provides tags of reaction type for each reaction data, we didn't use them, following the previous retro-reaction prediction publications.

## Implementation details

We employed the architecture of 6 BERT$_{base}$ encoder layers for our PV encoder and SMILES encoder, and 6 BERT$_{base}$ encoder layers with cross-attention layers for our fusion encoder. With given $\mathbf{Q} \in \mathbb{R}^{len_q \times d_k}$, $\mathbf{K} \in \mathbb{R}^{len_k \times d_k}$, and $\mathbf{V} \in \mathbb{R}^{len_k \times d_v}$ as query, key, and value inputs, the self-attention and cross-attention layers in BERT compute the output of the scaled-dot attention according to the following formula:

$$Attention(\mathbf{Q}, \mathbf{K}, \mathbf{V}) = Softmax\left(\frac{\mathbf{Q}\mathbf{K}^T}{\sqrt{d_k}}\right)\mathbf{V} \quad (17)$$

We pre-trained the model until it converges using a batch size of 96 and the AdamW optimizer with a weight decay of 0.02. The learning rate is warmed up to $10^{-4}$ and decreased to $10^{-5}$ with a cosine scheduler. We used the momentum-adjusting hyperparameter $\alpha$ of 0.4. Since the pseudo-label from the momentum teacher is not useful in the early stages of the training, we linearly increased $\alpha$ from 0 to 0.4 during the first epoch. The EMA hyperparameter $\lambda$ was fixed to 0.995, and the size of the PV and SMILES queue $k$ was set to 24,576. The momentum models are not used for downstream tasks. The pre-training was done

with 8 Nvidia A100 GPUs for about 52,000 batch iterations, which took roughly 12 h. The full description of training for downstream tasks is in Supplementary Table 6.

## Reporting summary

Further information on research design is available in the Nature Portfolio Reporting Summary linked to this article.

## Data availability

The pre-training dataset is publicly available in PubChem https://pubchem.ncbi.nlm.nih.gov/. ZINC15 test molecules for SMILES-to-PV generation are accessible through the ZINC15 website. Scaffold-split MoleculeNet datasets are available via DeepChem python module https://deepchem.io/, and raw databases can be found in the MoleculeNet website https://moleculenet.org/. The DILI training and test data preparation can be found in https://pubmed.ncbi.nlm.nih.gov/29788510/. The USPTO-480k and USPTO-50k dataset is available at https://github.com/wengong-jin/nips17-rexgen and https://github.com/MolecularAI/Chemformer/tree/main. Source data are provided with this paper.

## Code availability

The source code for SPMM, a list of 53 properties for PV, experimental codes, and datasets are available at https://github.com/jinhojsk515/SPMM/(DOI: 10.5281/zenodo.10567599)[66] to allow reproducing the results.

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

## Acknowledgements

This work was supported by the Institute of Information & communica-tions Technology Planning & Evaluation (IITP) grant funded by the Korea government (MSIT) (No.2019-0-00075, Artificial Intelligence Graduate School Program (KAIST)), National Research Foundation (NRF) of Korea grant NRF-2020R1A2B5B03001980 and RS-2023-00262527, and by the KAIST Key Research Institute (Interdisciplinary Research Group) Project.

## Author contributions

J.C. prepared the code, performed all experiments and analyses, col-lected data, and wrote the manuscript. J.C.Y. supervised the project in conception and discussion, and prepared the manuscript.

## Competing interests

The authors declare no competing interests.
