## [Peer Review File · Nature Communications]

Bidirectional Generation of Structure and Properties Through a Single Molecular Foundation ModelREVIEWER COMMENTS

Reviewer #1 (Remarks to the Author):

I am writing to provide my comments on the article titled "Bidirectional Generation of Structure and Properties Through a Single Molecular Foundation Model" by the authors. The article presents a bidirectional molecular foundation model that can be used for both molecular structure and property inferences. The authors claim that they achieve state-of-the-art performance and interpretable attention maps in both multimodal and unimodal tasks, including conditional molecule generation, property prediction, molecule classification, and reaction prediction. In order to be considered further for publication, the authors need to address the following comments:

Major comments:

1. The validity, novelty, and uniqueness of each molecule generation experiment should be posted in a table or picture in the article. These results can demonstrate the molecule generation ability of the model, and the authors should provide clear analysis of them.

2. The authors state that "SPMM had similar performance with other state-of-the-art reaction prediction models, although the number of the training data size of our model (i.e. 10M molecules) is much smaller than 100M molecules in Chemformer." However, the gap between Chemformer and SPMM is too large, which reaches 4.2%. While this work focuses on the generality of the model in various downstream tasks, it is essential to compare the performance between Chemformer and SPMM when pre-trained on 100M molecules.

3. The Chemformer can make retro-reaction predictions. Can SPMM perform this task?

4. Top-k is the currently accepted metric for the reaction prediction field. It is important because the reader needs to know if the models are predicting the correct structure but not in the top-1 or if they are failing altogether. The authors should show not just top-1 but also top-k, as Ref 4 did.

5. Benchmark results on MoleculeNet downstream tasks are not enough. The authors should update the results from the work of Molformer (Large-Scale Chemical Language Representations Capture Molecular Structure and Properties). It seems that SPMM does not achieve comparable performance compared to Molformer in MoleculeNet downstream tasks.

Minor comments:

1. What is the effect of pretraining set size? Is there any guidance on how small a pretraining set one could use to effectively employ this strategy?

2. Chemists tend to use compound identifiers (usually numbered starting at 1 from the first structure) to refer to a structure in the images. The structure format should be Kekule. This is the accepted method and would greatly improve readability.

3. The authors should upload a model demo and detailed datasets to GitHub or the web for reproduction.

4. The reason why the authors used the x-transformer rather than the vanilla Transformer should be added to the introduction. 5. The authors fed 1,000 SMILES that are not contained in the pre-training dataset to the pre-trained SPMM and generated their corresponding PV. To demonstrate the correlation between those molecules and pretraining datasets, the authors should randomly sample examples from them to visualize, using tools like t-SNE or PCA.

Thank you for considering my comments. I look forward to the authors' response to these concerns.

Reviewer #2 (Remarks to the Author):

This paper proposes a multimodal-inspired molecular foundation model for bidirectional generation of molecular structure and properties. The proposed method is motivated by Vision-Language Pre-training (VLP), instead of image and text, the authors take molecular SMILES and property vector (PV) as the two input data types of input. Then two corresponding encoders are utilized to obtain the embeddings, followed by weight-sharing fusion encoders for information flow. A contrastive loss is employed to predict whether the SMILES embeddings and property embeddings are from the same molecule. The trained model is then utilized for various possible downstream tasks, i.e., SMILES-to-PV generation, PV-to-SMILES generation, property prediction, reaction prediction. Multiple experiments are conducted to evaluate the performance of the proposed method. The intuition of developing a universal foundation model for various molecule-related tasks is promising.

- The authors propose to utilize 53 molecular properties generated from RDKit as another data modal for the pretraining. However, there is no information to prove or explain why these properties could be helpful for learning a generalized molecular representation. How exactly are these properties selected? What are the criteria for selecting the properties? How to ensure the selected properties have certain correlations with SMILES?
- One essential requirement of Language models is that the order of input data is meaningful, such as a sentence. A molecular SMILES is suitable for NLP models since SMILES representation is defined in an ordered manner. In the proposed framework, what is the order of the property information? It seems that the authors manually preset the order of these properties, and do not provide any explanations. It is questionable to naively form such a data source for NLP models and perform next property prediction. If the order is meaningless and arbitrary, the design of employing Transformer to predict the next property is reckless and unjustified.
- The authors attempt to propose a novel foundation model for molecules with self-supervised learning method, while the strategies are masking and next object (word or property) prediction, which are standard and fundamental techniques in self-supervised learning. The technical novelty is trivial.
- Several important related works and baselines are not introduced before discussing the details of the proposed method, e.g., x-attention Transformer architecture, dual-stream VLP model, cross-attention, and contrastive learning, which makes the paper difficult to follow.
- The proposed method is not clearly introduced. Many aspects are simply mentioned, but not explained how and why they work. For example, how does the fusion happens between modal? There should be some illustrations to describe the underlying computation, such as a formal presentation that includes the two embedding with their dimensions specified. Moreover, the authors state that SMILES is processed through fragment tokens, but there is no description of it. The utilized BPE algorithm is also not introduced and has no reference. Also, the authors utilize the Momentum teacher and EMA update, but do not give any details. Figure 1 is confusing with messy information but limited descriptions. Such vague presentations happen throughout the entire manuscript.
- In P.18, choosing negative pairs by hard-negative mining is a good way to distinguish similar molecules. However, with the masking action in PV, it is possible that PV is falsely selected as negative pair with the high masking ratio condition. How to avoid such problems?
- The experimental results are not convincing and fairly justified. The authors have conducted multiple experiments to demonstrate that their method can be utilized in various downstream tasks, and performed further analysis to illustrate the effectiveness such as attention visualization. However, the results do not seem to be significant, and the validation criteria are confusing. For

Table 1, the authors mention that the experiments are based on the average of three runs with different random seeds, but they do not report standard deviation. They also take the baseline results from ChemRL-GEM and ChemBERTa-2, but the experimental settings of these two models are different. ChemRL-GEM is based on 4 runs, while ChemBERTa-2 seems only run one time. The authors should compare all the baseline models following the exact experimental setting as their proposed method to ensure a fair comparison. For Table 2, the performance of the proposed SPMM is the second worst, which hardly demonstrates its superiority.

- For the SMILES-to-PV and PV-to-SMILES generation, the validity cannot be verified. The authors neither perform wet-lab experiments to veritably test the generated property values, nor prove the generated SMILES are practically valid with details. The designed evaluation criteria lack authenticity.
- The authors claim that the proposed model is generalized, but the method of offering certain additional property information without reasonable justification cannot be considered to be universal.

Minors:

- The introduction of foundation models is confusing and seems unnecessary. There is no clear statement that why the authors introduce such a term instead of commonly used pre-train models. The statement "Specifically, foundation models refer to the neural network models that are pre-trained in a self-supervised manner with a wide range of data" is also incorrect. Foundation models are generally trained in a self-supervised manner, but are not required [14].
- Several essential references are missing, e.g., the masked language model in P.2, VLP applications mentioned in P.3, and the meaningful tasks for cross-modal comprehension.
- In P.18, the terms in the equations are unclear, i.e., in Eq. (2) and (3), what are "m" and "n" standing for in the numerator?

[14]. Bommasani, R. et al. On the opportunities and risks of foundation models (2021).

Reviewer #3 (Remarks to the Author):

The manuscript by Chang et al "Bidirectional Generation of Structure and Properties Through a Single Molecular Foundation Model" describes a bidirectional molecular foundation model that can be used for both molecular structure and property inferences through a single model. The manuscript is well written.

Minor:

The author describe "property conditioned molecule generation (properties-to-SMILES)" (Fig. 1b), which is the task of inverse QSA(P)R. Please refer to "inverse QSAR" e.g. in the introduction and cite e.g.

- Kotsias, PC., et al. Direct steering of de novo molecular generation with descriptor conditional recurrent neural networks. *Nat Mach Intell* 2, 254–265 (2020).
- Bort W., et al. Inverse QSAR: Reversing Descriptor-Driven Prediction Pipeline Using Attention-Based Conditional Variational Autoencoder, *Journal of Chemical Information and Modeling* 2022 62 (22), 5471-5484

Please cite the original publications in the Introduction around citation 7:

- Gómez-Bombarelli, R.; Wei, J.; Duvenaud, D.; Hernández-Lobato, J.; Sánchez-Lengeling, B.; Sheberla, D.; Aguilera-Iparraguirre, J.; Hirzel, T.; Adams, R.; Aspuru-Guzik, A. Automatic Chemical Design Using a Data-Driven Continuous Representation of Molecules. *ACS Cent.*

Sci. 2018, 4, 268– 276

- Segler, M. H. S.; Kogej, T.; Tyrchan, C.; Waller, M. P. Generating Focused Molecule Libraries for Drug Discovery with Recurrent Neural Networks. ACS Cent. Sci. 2018, 4, 120– 131,
- Olivecrona, M.; Blaschke, T.; Engkvist, O.; Chen, H. Molecular De Novo Design through Deep Reinforcement Learning. J. Cheminf. 2017, 9, 48

P5 "controllable properties wasn't very large." What is not "very large"? Generative models do not have in theory a limit in properties taking into account for scoring & learning. Scoring functions for generative models (e.g. RNN) with 100s of properties are described, e.g. of target panels like kinases (~450 in the human genome). This is also true for property prediction models e.g. kinase panel activity.

Further your input vector is also fixed with 53 properties? The approach resembles multi-task learning, without mentioning it - maybe introduce a sentence in the e.g. introduction? This part should be re-written.

Please briefly explain (and cite origin) [CLS] token in methods.

- BERT: Pre-training of Deep Bidirectional Transformers for Language Understanding
arXiv:1810.04805

P6 "mean normalized RMSE of 0.2216" Which digit is significant/relevant - are four digits needed for significance/accuracy? ->similar question related to Fig2

P12 "study on 8" rule of convention is that numbers smaller than 10 are spelled out -> study on eight

P17 "in BERT ([c])." ([c])? "Encoding ([a])"? "Unknown ([b])"? - missing citations?

Major:

P12 How does SPMM handle noise, non-linearity etc.? The eight Molecule Net tasks are limited, it would be favorable if these could be extended to more complex endpoints. E.g. there are large annotated DILI datasets publicly available, such as FDA LTKB and NIH LiverTox.

- Gogishvili D. et al. Nonadditivity in public and inhouse data: implications for drug design. J Cheminform 13, 47 (2021).
- Kwapien K. et al, Implications of Additivity and Nonadditivity for Machine Learning and Deep Learning Models in Drug Design, ACS Omega 2022 7 (30)
- Dablander M., Exploring QSAR Models for Activity-Cliff Prediction, <https://arxiv.org/abs/2301.13644>van
- Van Tilborg D. et al., Exposing the Limitations of Molecular Machine Learning with Activity Cliffs, Journal of Chemical Information and Modeling 2022 62 (23), 5938-5951

NCOMMS-23-06428-T

Reply to the Reviewers

General Comments

We thank the editor and the reviewers for the constructive reviews. Your comments have helped us a lot in improving the quality of manuscript. We would first like to list the major changes that we have made to the manuscript.

1. We pre-trained our SPMM with bigger model architecture and more training data, improving the overall multimodal and unimodal performance of SPMM. We also adopted a new technique of SMILES augmentation [1] for forward and retro-reaction prediction tasks, which also increased the accuracy.
2. Downstream task study on retro-reaction prediction task and DILI classification task was added. The retro-reaction prediction task was done very similar to the forward prediction task but with the swap of reactants and products, and SPMM showed comparable results to state-of-the-art models in the USPTO-50k dataset [2]. The DILI prediction task was done to overcome the potential limitation of open databases and to verify if SPMM could be extended to more complex endpoints, in which SPMM outperformed the former suggested model.
3. Per reviewers' comments, we've slightly modified our results on SMILES-to-PV generation task and MoleculeNet downstream tasks: SMILES-to-PV generation task was newly done with 1,000 unseen molecules from ZINC15 database [4] to test the model with more out-of-distribution samples. MoleculeNet benchmark result was extended with the performance of Molformer [24].
4. We enriched the overall descriptions about our proposed model architecture and training objectives to make the manuscript easier to follow and justify our approach.

With the general modifications mentioned above, we have conducted more experiments and made proper modifications to the manuscript. For point-to-point response, please see below.

Reply to the Reviewer1

Comments to author: I am writing to provide my comments on the article titled "Bidirectional Generation of Structure and Properties Through a Single Molecular Foundation Model" by the authors. The article presents a bidirectional molecular foundation model that can be used for both molecular structure and property inferences. The authors claim that they achieve state-of-the-art performance and interpretable attention maps in both multimodal and unimodal tasks, including conditional molecule generation, property prediction, molecule classification, and reaction prediction. In order to be considered further for publication, the authors need to address the following comments:

We thank you for your efforts to review and comment our manuscript. We have made our best efforts to clarify the details of the proposed method and back up our claims. Please refer to our replies below.

R1.C1 The validity, novelty, and uniqueness of each molecule generation experiment should be posted in a table or picture in the article. These results can demonstrate the molecule generation ability of the model, and the authors should provide clear analysis of them.

Done. We now added quantitative results including validity, uniqueness, and novelty for our molecule generation experiments.

R1.C2. The authors state that "SPMM had similar performance with other state-of-the-art reaction prediction models, although the number of the training data size of our model (i.e.10M molecules) is much smaller than 100M molecules in Chemformer." However, the gap between Chemformer and SPMM is too large, which reaches 4.2%. While this work focuses on the generality of the model in various downstream tasks, it is essential to compare the performance between Chemformer and SPMM when pre-trained on 100M molecules.

According to the raised limitation suggestions, we increased the model size and the size of the pre-training data to 20 million. With this scale-up and introduction of a new technique of SMILES augmentation [1], SPMM now achieves the best top-1 accuracy on the forward-reaction prediction task and outperforms many other models.

R1.C3. The Chemformer can make retro-reaction predictions. Can SPMM perform this task?

Yes. Similar to the forward-reaction prediction task, we can utilize SPMM for the retro-reaction prediction task by taking a product and generating reactants in an autoregressive manner. We fine-tuned SPMM to retro-reaction prediction task for the USPTO-50k dataset [2] (following Chemformer and many previous works), and SPMM showed comparable results to Chemformer and other state-of-the-art models. We added the results of the retro-reaction task in USPTO-50k in the manuscript.

R1.C4. Top-k is the currently accepted metric for the reaction prediction field. It is important because the reader needs to know if the models are predicting the correct structure but not in the top-1 or if they are failing altogether. The authors should show not just top-1 but also top-k, as Ref 4 did.

We extended our analysis of the forward-reaction and retro-reaction prediction task to top-k accuracy and listed its result in the revised manuscript.

R1.C5. Benchmark results on MoleculeNet downstream tasks are not enough. The authors should update the results from the work of Molformer (Large-Scale Chemical Language Representations Capture Molecular Structure and Properties). It seems that SPMM does not achieve comparable performance compared to Molformer in MoleculeNet downstream tasks.

We found that in some MoleculeNet downstream tasks, the work of Molformer had used a different dataset preparation process from ours and the other models we had compared. We used a scaffold splitter algorithm for all tasks to split the dataset into train/validation/test sets in a ratio of 8:1:1 whereas the Molformer used random splitting for their MoleculeNet experiments. Therefore, we took the code and the model checkpoint of Molformer from their official GitHub to run MoleculeNet experiments with our data preparation.

We updated their performance in our results, and the overall performance of SPMM is still comparable to state-of-the-art models including Molformer.

R1.C6. What is the effect of pretraining set size? Is there any guidance on how small a pretraining set one could use to effectively employ this strategy?

In general, the performance of the model improves as the size of the pre-training dataset is larger. That said, we've experimented and observed during the development that the model also works effectively on multimodal tasks with pre-training using only one million molecules, which is 5% of the pre-training data for the main results.

R1.C7. Chemists tend to use compound identifiers (usually numbered starting at 1 from the first structure) to refer to a structure in the images. The structure format should be Kekule. This is the accepted method and would greatly improve readability.

Thank you for the constructive feedback. We've changed our visualization format of molecular structures for the manuscript, with Kekule form and additional numeric identifiers.

R1.C8. The authors should upload a model demo and detailed datasets to GitHub or the web for reproduction.

We uploaded our code, model checkpoints, and demo jupyter notebooks in the following GitHub link: <https://github.com/jinhojsk515/SPMM>.

R1.C9. The reason why the authors used the x-transformer rather than the vanilla Transformer should be added to the introduction.

The term "x-transformer" or "x-attention" refers to the cross-attention mechanism of the vanilla transformer architecture in SPMM. We apologize for the confusion, and for clarity, we added a more detailed explanation of the concept of cross-attention in the introduction.

R1.C10. The authors fed 1,000 SMILES that are not contained in the pre-training dataset to the pre-trained SPMM and generated their corresponding PV. To demonstrate the correlation between those molecules and pretraining datasets, the authors should randomly sample examples from them to visualize, using tools like t-SNE or PCA.

We took 1,000 SMILES that we used for the PV generation task and 5,000 sampled SMILES from the pre-training dataset to run UMAP [23] with their Morgan Fingerprint. UMAP is a dimensionality reduction algorithm similar to t-SNE. We found that even though the two SMILES sets are disjoint, the distribution of the 1,000 tested molecules mostly overlaps with the pre-training dataset. See Figure below.

Although we believe this is a relatively natural behavior since our 20 million SMILES from PubChem covers a very wide and general range of molecules, we decided to prepare another set of 1,000 SMILES from ZINC15 [4]. These are relatively out-of-distribution samples from our pre-training dataset as shown in the figure above, and the SPMM still generated correct PVs with respect to them. We changed our PV generation results to the results from 1,000 ZINC15 molecules.

Reply to the Reviewer2

Comments to author: This paper proposes a multimodal-inspired molecular foundation model for bidirectional generation of molecular structure and properties. The proposed method is motivated by Vision-Language Pre-training (VLP), instead of image and text, the authors take molecular SMILES and property vector (PV) as the two input data types of input. Then two corresponding encoders are utilized to obtain the embeddings, followed by weight-sharing fusion encoders for information flow. A contrastive loss is employed to predict whether the SMILES embeddings and property embeddings are from the same molecule. The trained model is then utilized for various possible downstream tasks, i.e., SMILES-to-PV generation, PV-to-SMILES generation, property prediction, reaction prediction. Multiple experiments are conducted to evaluate the performance of the proposed method. The intuition of developing a universal foundation model for various molecule-related tasks is promising.

Thank you for your thoughtful comments. Please see our point-to-point response below.

R2.C1. The authors propose to utilize 53 molecular properties generated from RDKit as another data modal for the pretraining. However, there is no information to prove or explain why these properties could be helpful for learning a generalized molecular representation. How exactly are these properties selected? What are the criteria for selecting the properties? How to ensure the selected properties have certain correlations with SMILES?

For the selection of properties for an informative PV modality, we decided to use as many basic properties as we could that can be easily calculated, rather than manually picking out properties through specific human-driven prior knowledge. Specifically, we harvested more than 120 features from the *RDKit Descriptor* Python module and ruled out properties that return an error, NaN(Not a Number), or a blank value for some molecules.

In this work, we treated PV with these property values as an independent modality separate from SMILES. As long as the information they contain expresses the source molecule correctly, there is no need for explicit correlation with the SMILES expression (although we're fairly confident that most of our PV properties have a strong relationship with molecular structure, as SPMM learned in Figure 5). In fact, providing richer information that SMILES doesn't contain enables the enhancement of molecule representations, the main purpose of introducing multimodality.

R2.C2. One essential requirement of Language models is that the order of input data is meaningful, such as a sentence. A molecular SMILES is suitable for NLP models since SMILES representation is defined in an ordered manner. In the proposed framework, what is the order of the property information? It seems that the authors manually preset the order of these properties, and do not provide any explanations. It is questionable to naively form such a data source for NLP models and perform next property prediction. If the order is meaningless and arbitrary, the design of employing Transformer to predict the next property is reckless and unjustified.

We understand that the concept of Next Property Prediction (NPP) using pre-defined order can be confusing. However, autoregressive modeling is not a training paradigm only possible for sequential data like language. In the image domain, models like MADE [5], pixelCNN [6], and DALL-E [7] demonstrated that images can be generated autoregressively by giving a pre-defined order to pixels or

image patches. You et al. have built a graph generation model called graphRNN [8] by treating graphs as a sequence of its nodes and edges and training RNN in an autoregressive manner to learn the joint distributions of edges and nodes.

The loss of Next Property Prediction is on the same track of treating data sequentially with pre-defining orders. It's true that the order in our PV is arbitrary, but the order is fixed so that each element's position has a meaning of the corresponding property. Since there are certain correlations between the elements in the PV as we mentioned in the manuscript, autoregressive modeling is also one of the possible choices for learning the distribution of PV.

R2.C3. The authors attempt to propose a novel foundation model for molecules with self-supervised learning method, while the strategies are masking and next object (word or property) prediction, which are standard and fundamental techniques in self-supervised learning. The technical novelty is trivial.

As we stated in the manuscript, one of our technical contributions is to demonstrate that each element in PV can be handled independently, like a word in a sentence, rather than treating PV as a single vector with properties like many existing works. This insight enabled us to apply recent multimodal model architectures and perform the transformer's attention mechanism to each element of a PV.

In the process of integrating multimodal learning models into our structure-property chemical domain, various novel ideas like autoregressive PV modeling and 50% random property masking were introduced. Furthermore, the combination of the various techniques in self-supervised learning and VLP for bidirectional chemical structure and property generation have never been tried before, which should not be regarded as trivial.

R2.C4. Several important related works and baselines are not introduced before discussing the details of the proposed method, e.g., x-attention Transformer architecture, dual-stream VLP model, cross-attention, and contrastive learning, which makes the paper difficult to follow.

By accepting the feedback, we attributed more explanation and related works in the introduction to help readers easy to follow. The term "x-attention" referred to the "cross-attention" mechanism of the transformer [3] architecture, which is now mentioned in the introduction paragraph of VLP. The concept of dual-stream VLP model is explained with the overall model description of SPMM.

R2.C5. The proposed method is not clearly introduced. Many aspects are simply mentioned, but not explained how and why they work. For example, how does the fusion happens between modal? There should be some illustrations to describe the underlying computation, such as a formal presentation that includes the two embedding with their dimensions specified. Moreover, the authors state that SMILES is processed through fragment tokens, but there is no description of it. The utilized BPE algorithm is also not introduced and has no reference. Also, the authors utilize the Momentum teacher and EMA update, but do not give any details. Figure 1 is confusing with messy information but limited descriptions. Such vague presentations happen throughout the entire manuscript.

We'd like to apologize that you felt the explanation in this work was insufficient.

Our fusion encoder mixes the features from different modalities by a cross-attention mechanism, whose detailed process is equal to that in the vanilla transformer. In the first submitted manuscript, we implicitly expressed our self-attention and cross-attention mechanism by describing that we utilized BERT [9] layers. However, we now more clearly state that the cross-modal information exchange was done by an attention mechanism, whose formula was described in the implementation details in the method section. We provided more detailed descriptions of our SMILES tokenization and how we performed the BPE algorithm [10] to obtain the token dictionary.

The concept and the detailed formula for the momentum teacher and EMA update were described in the Methods section, so we added more references that utilized this technique [11, 12].

R2.C6. In P.18, choosing negative pairs by hard-negative mining is a good way to distinguish similar molecules. However, with the masking action in PV, it is possible that PV is falsely selected as negative pair with the high masking ratio condition. How to avoid such problems?

It's true that a PV from a different data pair that correctly describes the SMILES could be occasionally selected as a negative pair. This problem of misjudging matching instances to negative pairs also happens in vanilla vision-language tasks due to the noisy data curation or short/vague expressions of a natural text. Specifically, it has been observed that the matching loss as a pre-training objective can still correctly guide the model to learn contextual information if the occasion of mismatched inputs is small enough.

We also note that since the elements in PV are numerical values, false negative pairs happen more rarely in SPMM compared to vanilla VLP tasks. For instance, if "MW=[UNK], logP=2.2" were selected with hard-negative mining for benzene, although the value is very close to the true logP of 2.1 of benzene, it's still a negative pair compared to the true value.

R2.C7. The experimental results are not convincing and fairly justified. The authors have conducted multiple experiments to demonstrate that their method can be utilized in various downstream tasks, and performed further analysis to illustrate the effectiveness such as attention visualization. However, the results do not seem to be significant, and the validation criteria are confusing. For Table 1, the authors mention that the experiments are based on the average of three runs with different random seeds, but they do not report standard deviation. They also take the baseline results from ChemRL-GEM and ChemBERTa-2, but the experimental settings of these two models are different. ChemRL-GEM is based on 4 runs, while ChemBERTa-2 seems only run one time. The authors should compare all the baseline models following the exact experimental setting as their proposed method to ensure a fair comparison. For Table 2, the performance of the proposed SPMM is the second worst, which hardly demonstrates its superiority.

Following the experiment condition of ChemRL-GEM, we updated SPMM's performance for each MoleculeNet downstream with the metric's average and standard deviation of four random seeds. All benchmark results from ChemRL-GEM now include their standard deviation. In the case of ChemBERTa-2, we couldn't find model checkpoints or the standard deviation of their results. We mentioned this in our revised manuscripts. However, we think this doesn't harm the significance of this result since SPMM outperforms all of them, even with less room to get affected by cherry-picking of random fluctuations.

For the forward-reaction prediction, we increased the number of the parameters, used more molecules in the pre-training up to 20 million, and introduced a technique of SMILES augmentation for fine-tuning. SPMM now achieves the best top-1 accuracy on the forward-reaction prediction task.

R2.C8. For the SMILES-to-PV and PV-to-SMILES generation, the validity cannot be verified. The authors neither perform wet-lab experiments to veritably test the generated property values, nor prove the generated SMILES are practically valid with details. The designed evaluation criteria lack authenticity.

*for PV-to-SMILES generation

In many papers that explored *in silico* molecule generation tasks, the most widely adopted definition of "validity" is the proportion of chemically possible samples among the overall generated samples [13, 14]. In the case of SMILES notation, the validity would be the ratio of the number of generated strings

that follow the grammar of SMILES over the total generated strings. We described these validity metrics and also showed that the generated molecules actually agree with the input condition.

"Uniqueness" and "Novelty" are other widely used metrics along with validity for molecule generation models defined as follows:

$$\begin{aligned} \text{validity} &= \frac{\text{\#SMILES with valid syntax}}{\text{\#generated SMILES}} \\ \text{uniqueness} &= \frac{\text{\#nonduplicate valid SMILES}}{\text{\#valid SMILES}} \\ \text{novelty} &= \frac{\text{\#unique SMILES not in the pretraining data}}{\text{\#unique SMILES}} \end{aligned}$$

The revised manuscript provides the validity for each conditional molecule generation experiment and also describes metrics of uniqueness and novelty for more quantitative analysis.

*for SMILES-to-PV generation

We used RMSE for the evaluation metrics of SMILE-to-PV generation. The target label was the output of the Python module that we used to calculate properties to build PV for pre-training. These are the answers that the model was given for its pre-training, and it's fair to compare the generated PV with this to demonstrate what the model learned. Many papers on *de novo* molecular design [13, 14, 15, 16] and property prediction models [17] compared their model output to the computational module outputs or validation set labels in benchmark datasets like MoleculeNet, assuming they're the truth value. We followed this common approach and believe this can be justified without actual wet-lab experiments.

That said, we agree that the lack of wet lab experiments may be a limitation and need further study, so we added this point in the Discussion section.

R2.C9. The authors claim that the proposed model is generalized, but the method of offering certain additional property information without reasonable justification cannot be considered to be universal.

We would like to respond to this consideration in two respects.

Regarding the concerns about using additional information, we want to emphasize that all properties that we used can be easily computed. We believe that introducing additional data to learn more informative chemical representations is acceptable since there is no specific problem condition that explicitly limits the available data. Liu et al. have collected natural text descriptions for molecules online and employed them for multimodal learning [18]. The Morgan fingerprint [19], a conventional representation vector that many biochemical/pharmaceutical AI models have used [20], also contains additional structural information provided by the fingerprint construction rules.

If you're concerned about whether the chosen 53 properties are universally good for all downstream tasks, we want to emphasize that we used various easily-accessible properties as much as possible to build a broadly applicable representation, whose performance was shown in single-modality downstream tasks.

In addition, our main purpose is to present that a collection of properties can be a data modality not only than supervised labels, and multimodal learning using this provides a novel methodology for the better representation of chemical instances. The 53 properties used in this work are only an example of the success of this methodology by broadly prepared properties, and it can be generally and flexibly applicable by using a more appropriate choice of properties depending on the field.

R2.C10. The introduction of foundation models is confusing and seems unnecessary. There is no clear statement that why the authors introduce such a term instead of commonly used pre-train models. The statement “Specifically, foundation models refer to the neural network models that are pre-trained in a self-supervised manner with a wide range of data” is also incorrect. Foundation models are generally trained in a self-supervised manner, but are not required [14].

We agree that the original expression was inaccurate, and the expression should be “foundation models are *generally* trained with self-supervised learning”. Although we had composed the paragraph since it was the concept where we'd got the motivation, we accepted the suggestion and removed that part, briefly mentioning them in the introduction of pre-trained models.

R2.C11. Several essential references are missing, e.g., the masked language model in P.2, VLP applications mentioned in P.3, and the meaningful tasks for cross-modal comprehension.

The expression “masked language model” was a typo of “masked language modeling,” one of the two training objectives in BERT that we referenced in that sentence. For the VLP applications and chemical cross-modal comprehension tasks, we newly included a citation of specific benchmark challenges or several works related to them.

R2.C12. In P.18, the terms in the equations are unclear, i.e., in Eq. (2) and (3), what are “m” and “n” standing for in the numerator?

“M” and “N” represent the number of SMILES and PV participating in the contrastive loss calculation, which would be equal to the sum of the batch size and the queue size k . Including this description, we revised the equations for the training objectives for more clarity.

Reply to the Reviewer3

Comments to author: The manuscript by Chang et al "Bidirectional Generation of Structure and Properties Through a Single Molecular Foundation Model" describes a bidirectional molecular foundation model that can be used for both molecular structure and property inferences through a single model. The manuscript is well written.

We would like to appreciate the reviewer for encouraging and constructive feedback. Please kindly check our response provided below.

R3.C1. The author describe "property conditioned molecule generation (properties-to-SMILES)" (Fig. 1b), which is the task of inverse QSA(P)R. Please refer to "inverse QSAR" e.g. in the introduction and cite e.g.

- Kotsias, PC., et al. Direct steering of de novo molecular generation with descriptor conditional recurrent neural networks. Nat Mach Intell 2, 254–265 (2020).
- Bort W., et al. Inverse QSAR: Reversing Descriptor-Driven Prediction Pipeline Using Attention-Based Conditional Variational Autoencoder, Journal of Chemical Information and Modeling 2022 62 (22), 5471-5484

Thank you for the constructive suggestions. We've mentioned that conditional molecule generation is also referred as “inverse-QSAR” and cited the related work.

R3.C2. Please cite the original publications in the Introduction around citation 7:

- Gómez-Bombarelli, R.; Wei, J.; Duvenaud, D.; Hernández-Lobato, J.; Sánchez-Lengeling, B.; Sheberla, D.; Aguilera-Iparraguirre, J.; Hirzel, T.; Adams, R.; Aspuru-Guzik, A. Automatic Chemical Design Using a Data-Driven Continuous Representation of Molecules. *ACS Cent. Sci.* 2018, 4, 268–276
- Segler, M. H. S.; Kogej, T.; Tyrchan, C.; Waller, M. P. Generating Focused Molecule Libraries for Drug Discovery with Recurrent Neural Networks. *ACS Cent. Sci.* 2018, 4, 120–131,
- Olivecrona, M.; Blaschke, T.; Engkvist, O.; Chen, H. Molecular De Novo Design through Deep Reinforcement Learning. *J. Cheminf.* 2017, 9, 48

We've added citations for the listed papers, the major pioneering works on *de novo* molecular design.

R3.C3. P5 "controllable properties wasn't very large." What is not "very large"? Generative models do not have in theory a limit in properties taking into account for scoring & learning. Scoring functions for generative models (e.g. RNN) with 100s of properties are described, e.g. of target panels like kinases (~450 in the human genome). This is also true for property prediction models e.g. kinase panel activity. Further your input vector is also fixed with 53 properties? The approach resembles multi-task learning, without mentioning it - maybe introduce a sentence in the e.g. introduction? This part should be re-written.

Indeed, many generative models for conditional molecule generation for hundreds of different properties have been reported. However, to the best of our knowledge, the proposed models' simultaneous control for multiple conditions was limited, mostly less than ten. In contrast, SPMM could generate suitable molecules that simultaneously match more than 50 explicit input property conditions. In regard to the reviewer's comment on RNN with 100s of properties, we could not find a reference so your feedback on this matter will be appreciated for the subsequent revision.

It's true that the NPP loss of SPMM, which predicts the values of various properties, resembles multi-task learning [21]. However, taking a step from using the property values as targets, we utilize them as inputs for a separate data modality. We added these similarities and differences between SPMM and multi-task learning to the revised manuscript.

R3.C4. Please briefly explain (and cite origin) [CLS] token in methods.

- BERT: Pre-training of Deep Bidirectional Transformers for Language Understanding arXiv:1810.04805

In the Method section, we added the explanation about what [CLS] token is and how its feature can be utilized as a representation vector for various downstream tasks.

R3.C5. P6 "mean normalized RMSE of 0.2216" Which digit is significant/relevant - are four digits needed for significance/accuracy? ->similar question related to Fig2

For the results in Table 1, we used three significant digits to follow the other benchmark results. However, for the RMSE you mentioned and the property values of Figure 2, rounding was done at an arbitrary number of digits. These values are simply model outputs, not measured values with certain margins of error. Therefore, we fixed inconsistent rounding in the script to depict three significant figures (to follow Table 1) for all metrics, except for Figure 2 to depict four significant figures to accurately show large properties like Molecular Weight. Thank you for kindly pointing it out.

R3.C6. "study on 8" rule of convention is that numbers smaller than 10 are spelled out -> study on eight

We fixed this issue. Thank you for the correction.

R3.C7. "in BERT ([c])." ([c])? "Encoding ([a])"? "Unknown ([b])"? - missing citations

The expression “[a]”, “[b]”, and “[c]” on page 17 were referring the subsections of Figure 6. We apologize for unclear wording with square brackets, which could give you the impression of missing citations. For consistency with Figure 1, we changed the square brackets in the script and Figure 6 into parentheses.

R3.C8. P12 How does SPMM handle noise, non-linearity etc.? The eight Molecule Net tasks are limited, it would be favorable if these could be extended to more complex endpoints. E.g. there are large annotated DILI datasets publicly available, such as FDA LTKB and NIH LiverTox.

- Gogishvili D. et al. Nonadditivity in public and inhouse data: implications for drug design. *J Cheminform* 13, 47 (2021).
- Kwapien K. et al, Implications of Additivity and Nonadditivity for Machine Learning and Deep Learning Models in Drug Design, *ACS Omega* 2022 7 (30)
- Dablander M., Exploring QSAR Models for Activity-Cliff Prediction, <https://arxiv.org/abs/2301.13644>van
- Van Tilborg D. et al., Exposing the Limitations of Molecular Machine Learning with Activity Cliffs, *Journal of Chemical Information and Modeling* 2022 62 (23), 5938-5951

We decided to verify our unimodal downstream task ability under the MoleculeNet benchmark because it was widely used in many works and clearly provides the separation of train/validation/test set for fair performance comparison. We believe that MoleculeNet downstream tasks are sufficiently noisy and non-linear as their labels are obtained through actual experiments, and the fact that SPMM achieved near-SOTA performance on these datasets demonstrates the generalization ability of SPMM to a variety of noisy non-linear tasks.

Nevertheless, to resolve the raised concerns, we ran additional experiments of fine-tuning SPMM on the Drug-Induced Liver Injury (DILI) prediction tasks, as the reviewer suggested. Since many works about the DILI machine learning model have built their own dataset rather than using widely-used benchmarks, we've found a DILI dataset preparation from a known publication [22] and fine-tuned SPMM with them for a fair comparison. For both datasets, SPMM outperformed the model from the previous work that provided the dataset. We added this result to the manuscript.

References

- [1] Bjerrum, E. Smiles enumeration as data augmentation for neural network modeling of molecules (2017).
- [2] Schneider, N., Stiefl, N. & Landrum, G. A. What's what: The (nearly) definitive guide to reaction role assignment 56, 2336–2346 (2016).
- [3] Vaswani, A. *et al.* Attention is all you need (2017).
- [4] Sterling, T. & Irwin, J. J. Zinc 15 – ligand discovery for everyone. *Journal of Chemical Information and Modeling* 55, 2324–2337 (2015).
- [5] Germain, M., Gregor, K., Murray, I. & Larochelle, H. Made: Masked autoencoder for distribution estimation. *arXiv:1502.03509 [cs, stat]* (2015).

- [6] Oord, A. v. d. *et al.* Conditional image generation with pixelcnn decoders. *arXiv:1606.05328 [cs]* (2016).
- [7] Ramesh, A. *et al.* Zero-shot text-to-image generation. *arXiv:2102.12092 [cs]* (2021).
- [8] You, J., Ying, R., Ren, X., Hamilton, W. L. & Leskovec, J. Graphrnn: Generating realistic graphs with deep auto-regressive models (2018).
- [9] Devlin, J., Chang, M.-W., Lee, K. & Toutanova, K. Bert: Pre-training of deep bidirectional transformers for language understanding (2018).
- [10] Gage, P. A new algorithm for data compression (1994).
- [11] Li, J. *et al.* Align before fuse: Vision and language representation learning with momentum distillation (2021).
- [12] Yu, J. *et al.* Coca: Contrastive captioners are image-text foundation models (2022).
- [13] Lim, J., Hwang, S.-Y., Moon, S., Kim, S. & Kim, W. Y. Scaffold-based molecular design with a graph generative model. *Chem. Sci.* 11, 1153–1164 (2020).
- [14] Kwon, K., Jung, K., Park, J., Na, H. & Shin, J. String-based molecule generation via multidecoder vae (2022).
- [15] Zhou, Z., Kearnes, S., Li, L., Zare, R. N. & Riley, P. Optimization of molecules via deep reinforcement learning (2019).
- [16] Lim, J., Ryu, S., Kim, J. W. & Kim, W. Y. Molecular generative model based on conditional variational autoencoder for de novo molecular design (2018).
- [17] Ryu, S. & Lee, S. Accurate, reliable and interpretable solubility prediction of druglike molecules with attention pooling and bayesian learning (2022).
- [18] Liu, S. *et al.* Multi-modal molecule structure-text model for text-based retrieval and editing (2022).
- [19] Morgan, H. L. The generation of a unique machine description for chemical structures—a technique developed at chemical abstracts service. *Journal of Chemical Documentation* 5, 107–113 (1965).
- [20] Capecchi, A., Probst, D. & Reymond, J.-L. One molecular fingerprint to rule them all: drugs, biomolecules, and the metabolome. *Journal of Cheminformatics* 12 (2020).
- [21] Crawshaw, M. Multi-task learning with deep neural networks: A survey (2020).
- [22] Ai, H. *et al.* Predicting drug-induced liver injury using ensemble learning methods and molecular fingerprints. *Toxicological Sciences* 165, 100–107 (2018).
- [23] McInnes, L., Healy, J. & Melville, J. Umap: Uniform manifold approximation and projection for dimension reduction (2018).
- [24] Ross, J. *et al.* Large-scale chemical language representations capture molecular structure and properties. *Nature Machine Intelligence* 4, 1256–1264 (2022).

REVIEWER COMMENTS

Reviewer #1 (Remarks to the Author):

I am writing to provide my comments on the article titled "Bidirectional Generation of Structure and Properties Through a Single Molecular Foundation Model" by the authors.

The revised manuscript of Bidirectional Generation of Structure and Properties Through a Single Molecular Foundation Model is a great improvement on the initial submission. The reviewer thanks the authors for their work on improving the study and the addition of the new models for better comparisons.

The manuscript is suggested to be published after the concerns below are:

1. The definitions of validity, novelty, and uniqueness should be included in the Method section instead of the Results section.
2. The validity, novelty, and uniqueness of each molecule generation experiment have been posted in a table of the article. However, the validity of model is only 0.751 ± 0.008 when the input has no property control, which seems not be a good sign.
3. The classic generation models such as GAN and VAE should be included in this paper as baselines for comparing SPMM in molecule generation tasks. Additionally, the validity, novelty, and uniqueness of those models should be presented in this paper.
4. The authors claim to have used data augmentation to enhance the model's capabilities. I'm interested in the impact of this method. In my experience, data augmentation is not always a reliable strategy and can sometimes yield worse results. Could you please provide the results of SPMM_no_aug (trained on 20 million data) in reaction prediction (both forward and retro) tasks? This will demonstrate the difference between SPMM_no_aug (trained on 20 million data) and SPMM_aug (trained on 20 million data with data augmentation).
5. As far as I know, the training time has an impact on the model's performance. Therefore, detailed experimental setting information such as the training time should be clearly provided in the manuscript/supplementary information (SI).

Thank you for considering my comments. I look forward to the authors' response to these concerns.

Reviewer #2 (Remarks to the Author):

Thank the authors for providing the responses and including more details in the manuscript. However, my main concerns are not sufficiently addressed. The intuition of designing a foundation model for molecular representation learning is good, and it is interesting that the authors seek another way to utilize molecular property data as another modality. However, the proposed method is not adequately designed and lacks the necessary theoretical support to substantiate the proposed statement. I would like to emphasize my main concerns again as follows:

- The property data has been manually selected by the authors in a predetermined order. The authors have argued that other modalities such as images and graphs also utilize autoregressive modeling. However, these types of data contain inherent meaning, i.e., images are arranged based on the order of pixels, and graphs have nodes connected by bonds. In contrast, the utilized PV does not possess underlying characteristics where each property is arbitrarily ranked. While the current design may yield comparable results by chance, there is no theoretical support to substantiate this claim.
- I understand that the authors may like to follow a straightforward implementation of the property modality, so that they utilize the next property prediction task. However, as

aforementioned, the order of the properties has no meaning, which makes such an approach inadequate.

- The authors have conducted many experiments, which is appreciated. However, the presented experiments primarily demonstrate the applicability of the proposed method to various tasks, which is expected since it is a molecular representation model. The performance results are insufficient to establish the solidity of SPMM. For example, in Figure 3, it is intuitive that the modification of the input would lead to changes in the output, there is a lack of quantitative evidence demonstrating a correlation between these changes. Similarly, in Figure 5, it could be subjective to pick two molecules as visualization. It is crucial for the authors to provide evidence that the attention scores genuinely reflect the underlying relationship between property and SMILES features, for example, conduct a statistical analysis involving all the fragments along with their corresponding attention values. In Table 2, since most of the baseline scores are obtained from GEM, the main comparison is between SPMM and GEM. However, as shown, GEM outperforms SPMM on 4/7 tasks, while SPMM only performs the best on 3/9 tasks. The performances can be considered comparable not competitive. Similar for Table 4, SPMM only performs the best for top-1 accuracy in the forward prediction task.

Overall, although the authors attempt to explore the utilization of molecular property data as an additional modality, the current approach falls short in its execution and does not provide sufficient evidence to support their claims. It is suggested that the authors conduct in-depth research regarding multi-modality learning with respect to tabular data type, which may help improve the methodology.

NCOMMS-23-06428A

Reply to the Reviewers

General Comments

We thank the editor and the reviewers for the constructive reviews. Your comments have helped us a lot in improving the quality of manuscript. We would first like to list the major changes that we have made to the manuscript.

1. Given that the performance difference may be originated from the small training data set, we further pre-trained our SPMM with more training data, improving the overall multimodal and unimodal performance of SPMM. We also adopted beam search for PV-to-SMILES generation tasks, same as reaction prediction tasks, which also increased the performance.
2. We have provided theoretical and empirical evidences to support our specific PV encoding and the reason of using PV as an additional modality. Specifically, the importance of the permutation invariance of positional encoding of Transformer, and the learnable positional embedding have been emphasized for better understanding of our motivation of using PV as additional modality.
3. Per reviewers' comments, we enriched the manuscript with more benchmark studies and additional analyses.

With the general modifications mentioned above, we have conducted more experiments and made proper modifications to the manuscript. For point-to-point response, please see below.

Reply to the Reviewer1

Comments to author: I am writing to provide my comments on the article titled "Bidirectional Generation of Structure and Properties Through a Single Molecular Foundation Model" by the authors. The revised manuscript of Bidirectional Generation of Structure and Properties Through a Single Molecular Foundation Model is a great improvement on the initial submission. The reviewer thanks the authors for their work on improving the study and the addition of the new models for better comparisons. The manuscript is suggested to be published after the concerns below. Thank you for considering my comments. I look forward to the authors' response to these concerns.

Thank you for your thoughtful comments. We have made our best efforts to clarify the details of the proposed method and back up our claims. Please refer to our replies below.

R1.C1 The definitions of validity, novelty, and uniqueness should be included in the Method section instead of the Results section.

We modified the manuscript according to the suggestion, moving the definitions of validity, novelty, and uniqueness to the Methods section.

R1.C2. The validity, novelty, and uniqueness of each molecule generation experiment have been posted in a table of the article. However, the validity of model is only 0.751 ± 0.008 when the input has no property control, which seems not be a good sign.

We thought that the situations without any given properties are not likely to happen, so we thought 75% validity was fine. That said, to improve the generated SMILES' validity per reviewer's comment, we adapted a beam search of k=2 instead of greedy sampling for PV-to-SMILES generation tasks. Although this slightly decreased the uniqueness and novelty as a trade-off, and the validity of the generated SMILES dramatically improved to more than 97% for every scenario we've tested. Thanks for the valuable comment.

R1.C3. The classic generation models such as GAN and VAE should be included in this paper as baselines for comparing SPMM in molecule generation tasks. Additionally, the validity, novelty, and uniqueness of those models should be presented in this paper.

Done. We have listed several conventional molecular generation models' validity, uniqueness, and novelty in the table below. The metrics for SPMM were obtained in unconditional generation and 53-property conditional generations. We've found that the SPMM showed higher validity, uniqueness, and novelty compared to the previous conventional generative models. We included this result in the manuscript.

model		Validity in %	Uniqueness in %	Novelty in %	V × U × N in %
unconditional	SMILES RNN ⁴⁰	93.2	-	89.9	83.8
	SMILES VAE ⁴⁰	80.4	-	79.3	63.8
	GraphVAE-NoGM ⁴¹	81.0	24.1	61.0	11.9
	molGAN ⁴²	98.1	10.4	94.2	9.61
	SPMM _{unconditional} *	97.1	99.1	95.0	91.4
conditional	scaffold-GGM ^{†10}	96.5	85.6	99.0	81.8
	SPMM _{deterministic} *	99.5	99.9	96.1	95.5
	SPMM _{stochastic} *	99.3	99.9	98.4	97.6

Table 2: Validity, uniqueness, novelty, and their accumulated product of the molecules generated by SPMM and other conditional/unconditional molecular generative models. The highest V × U × N value for unconditional models and conditional models are written in bold. The benchmark results were taken from molGAN⁴² and scaffold-GGM¹⁰. †The metrics are the mean of three separate models trained for different single-property conditions, namely logP, MW, and TPSA. * The SMILES generation was done with 1,000 unseen PVs, with all 53 properties are controlled for conditional generation.

R1.C4. The authors claim to have used data augmentation to enhance the model's capabilities. I'm interested in the impact of this method. In my experience, data augmentation is not always a reliable strategy and can sometimes yield worse results. Could you please provide the results of SPMM_{no_aug} (trained on 20 million data) in reaction prediction (both forward and retro) tasks? This will demonstrate the difference between SPMM_{no_aug} (trained on 20 million data) and SPMM_{aug} (trained on 20 million data with data augmentation).

References [a1] and [a2] previously reported that applying SMILES augmentation to reactant and product in a reaction prediction task improves the model performance, which results are confirmed by our experimental results with SPMM. Fine-tuning without SMILES augmentation decreased top-1 accuracy, and the result of this ablation study was added to the Supplementary results.

	forward acc.[↑]				retro acc.[↑]		
	top-1	top-2	top-3	top-5	top-1	top-5	top-10
augmentation X	0.879	0.914	0.929	0.938	0.445	0.597	0.626
augmentation O	0.915	0.935	0.946	0.954	0.534	0.676	0.703

Table 2. The ablation study of SMILES augmentation on the SPMM's forward and retro-reaction prediction task.

[a1] Tetko, I. V., Karpov, P., Van Deursen, R. & Godin, G. State-of-the-art augmented nlp transformer models for direct and single-step retrosynthesis. *Nature Communications* 11 (2020).

[a2] Irwin, R., Dimitriadis, S., He, J. & Bjerrum, E. J. Chemformer: a pre-trained transformer for computational chemistry. *Machine Learning: Science and Technology* 3, 015022 (2022)

However, as the reviewer pointed out from personal experience, this was not a universal solution that benefits every downstream task. [a2] reported that SMILES augmentation decreased the top-k ($k > 1$) result as a trade-off, which was also observed in our SPMM experiments. We also found that SMILES augmentation did not particularly improve performance on MoleculeNet and DILI downstream tasks. We suspect this is due to the fact that the nature of the task of sequence generation (e.g. reaction prediction) and sequence comprehension (e.g. property prediction) are different.

R1.C5. As far as I know, the training time has an impact on the model's performance. Therefore, detailed experimental setting information such as the training time should be clearly provided in the manuscript/supplementary information (SI).

Following the provided suggestions, we added more detailed information about pre-training in the Implementation details, including required training time and the number of training iterations.

Reply to the Reviewer2

Comments to author: Thank the authors for providing the responses and including more details in the manuscript. However, my main concerns are not sufficiently addressed. The intuition of designing a foundation model for molecular representation learning is good, and it is interesting that the authors seek another way to utilize molecular property data as another modality. However, the proposed method is not adequately designed and lacks the necessary theoretical support to substantiate the proposed statement. I would like to emphasize my main concerns again as follows:

Overall, although the authors attempt to explore the utilization of molecular property data as an additional modality, the current approach falls short in its execution and does not provide sufficient evidence to support their claims. It is suggested that the authors conduct in-depth research regarding multi-modality learning with respect to tabular data type, which may help improve the methodology.

We thank you for your efforts to review and comment on our manuscript. Please see our point-to-point response below.

R2.C1. The property data has been manually selected by the authors in a predetermined order. The authors have argued that other modalities such as images and graphs also utilize autoregressive modeling. However, these types of data contain inherent meaning, i.e., images are arranged based on the order of pixels, and graphs have nodes connected by bonds. In contrast, the utilized PV does not possess underlying characteristics where each property is arbitrarily ranked. While the current design may yield comparable results by chance, there is no theoretical support to substantiate this claim.

We would like to assure the reviewer that when there's no specific attention mask (e.g. causal attention mask as in GPT-4), the output of the transformer's self-attention and cross-attention mechanism is invariant to the position of each feature vector. This is thanks to the learnable positional embedding used in our Transformer architecture. For example, if the order of PVs is permuted in a different order, then the learnable positional embedding would learn different embedding that take into account the optimal alignment between the properties and the SMILES embeddings.

In fact, the permutational invariance of the positional encoding in Transformer architecture has been well documented in machine learning literature [a3]:

[a3] M. M. Naseer, K. Ranasinghe, S. H. Khan, M. Hayat, F. Shahbaz Khan, and M.-H. Yang, “Intriguing properties of vision transformers,” *Advances in Neural Information Processing Systems (NeurIPS)*, vol. 34, pp. 23 296–23 308, 2021

Furthermore, this permutational invariance of the positional encoding was the key that has been utilized to keep the privacy during the federated training of Transformer-based foundation model [a4]:

[a4] Park, S., Lee, I. J., Kim, J. W., & Ye, J. C. (2023). MS-DINO: Efficient Distributed Training of Vision Transformer Foundation Model in Medical Domain through Masked Sampling. *arXiv preprint arXiv:2301.02064*

In terms of pretraining objectives, the contrastive learning, PV-SMILES matching, and next SMILES token prediction treat all elements in a PV equally and non-sequentially. The only pre-training objective that the predetermined order matters is the next-property prediction which models the distribution of the PV themselves. However, many transformer-based generative models for tabular data [a5, a6] used next-entry prediction with a pre-determined order to learn the distribution of the tabular data, like ours.

[a5] Canale, L., Grislain, N., Lothe, G., & Leduc, J. (2022). Generative Modeling of Complex Data. *arXiv preprint arXiv:2202.02145*.

[a6] Castellon, R., Gopal, A., Bloniarz, B., & Rosenberg, D. (2023). DP-TBART: A Transformer-based Autoregressive Model for Differentially Private Tabular Data Generation. *arXiv preprint arXiv:2307.10430*.

Moreover, the recent theoretical work [a7] showed that autoregressive modeling is a universal learner that is not specific to any data type.

[a7] Malach, E. (2023). Auto-Regressive Next-Token Predictors are Universal Learners. *arXiv preprint arXiv:2309.06979*.

Given all this evidence, in contrast to reviewer’s concern, our way of utilizing the PV encoding is theoretically and empirically well-supported.

R2.C2. I understand that the authors may like to follow a straightforward implementation of the property modality, so that they utilize the next property prediction task. However, as aforementioned, the order of the properties has no meaning, which makes such an approach inadequate.

We would like to assure the reviewer that we did not follow a straightforward implementation of the property modality. Rather our approach is both theoretically and empirically well supported as demonstrated in large volume of machine learning literatures including [a3-a4] in the response to the previous comment.

Specifically, as we mentioned above, several transformer-based generative models that generate tabular data [a5, a6] use next-entry prediction with a pre-determined order like ours. One of those studies [a6] showed that reordering using criteria such as cardinality made no difference in performance.

In fact, we conducted an ablation study in which we pre-trained the model by replacing the next property prediction in the pretrain objective with a masked property prediction similar to BERT. In this case, position embedding is equivalent to column embedding in tabular data, which completely eliminates the predetermined order issue that the reviewer was concerned about. However, we found no significant

performance improvement in any of the downstream tasks. See Table S4 below, which we also added in the Supplementary Results.

PV-to-SMILES generation task										
model	sampling	input PV			Validity	Uniqueness	Novelty	normalized RMSE		
SPMM (NPP)	deterministic	1,000 unseen PubChem SMILES' PV			0.995 ±0.001	0.999 ±0.001	0.961 ±0.005	0.216 ±0.004		
	stochastic	full PV of the molecule 1 Molecular weight=150			0.974 ±0.005	0.905±0.007	0.998 ±0.003	0.185 ±0.004		
SPMM (MPM)	deterministic	1,000 unseen PubChem SMILES' PV			0.967±0.003	0.990±0.002	0.897±0.009	0.577±0.015		
	stochastic	full PV of the molecule 1 Molecular weight=150			0.956±0.006	0.999 ±0.001	0.987±0.003	0.625±0.021		
MoleculeNet		regression[RMSE, ↓]					classification[AUROC in %, ↑]			
Dataset	Delaney ESOL	LIPO	Freesolv	BACE	Clearance	BBBP	BACE	Clintox	SIDER	
SPMM (NPP)	0.817 ±0.010	0.681 ±0.004	1.868 ±0.041	1.041±0.022	42.607 ±0.675	75.1 ±0.9	84.4 ±0.4	92.7 ±0.7	66.9 ±0.9	
SPMM (MPM)	0.874±0.007	0.725±0.008	1.897±0.060	1.040 ±0.010	45.821±0.820	72.6±0.7	83.3±0.5	90.4±1.9	65.7±1.5	
DILI classification task		Acc in %[↑]	Selectivity in %[↑]	Specificity in %[↑]	AUROC in %[↑]					
SPMM (NPP)		84.4	83.9	84.6	92.6					
SPMM (MPM)		83.0	81.9	86.9	91.5					

Table S4: The ablation studies of replacing SPMM’s pre-training objective of NPP to masked property modeling (MPM), which aims to predict the properties that are replaced with the special masking token similar to BERT⁶¹. The better performance for each metric are written in bold.

R2.C3. The authors have conducted many experiments, which is appreciated. However, the presented experiments primarily demonstrate the applicability of the proposed method to various tasks, which is expected since it is a molecular representation model. The performance results are insufficient to establish the solidity of SPMM. For example, in Figure 3, it is intuitive that the modification of the input would lead to changes in the output, there is a lack of quantitative evidence demonstrating a correlation between these changes. Similarly, in Figure 5, it could be subjective to pick two molecules as visualization. It is crucial for the authors to provide evidence that the attention scores genuinely reflect the underlying relationship between property and SMILES features, for example, conduct a statistical analysis involving all the fragments along with their corresponding attention values. In Table 2, since most of the baseline scores are obtained from GEM, the main comparison is between SPMM and GEM. However, as shown, GEM outperforms SPMM on 4/7 tasks, while SPMM only performs the best on 3/9 tasks. The performances can be considered comparable not competitive. Similar for Table 4, SPMM only performs the best for top-1 accuracy in the forward prediction task.

Firstly, Figure 3 is a simple example of applying the PV-to-SMILES task to a practical molecule editing scenario. The reviewer is kindly reminded that SPMM’s ability to generate SMILES that exactly follow the input PV has already been explored in the previous PV-to-SMILES generation task results, including Table 1 and Figure 2.

For attention visualization examples including Figure 5, the attention pattern is not easy for quantitative analysis based on their absolute values since each fragment does not have a fixed explicit relationship to properties by itself (e.g., 'CCC' may or may not be part of a ring, depending on the surrounding fragments). That said, we observed which tokens show high attention scores for each property using 1,000 randomly sampled molecules’ cross-attention map. It is shown that certain property’s relatable tokens tend to show high attention score to that property; *TPSA* got tokens with polar atoms like oxygen and halogen atoms, *NumHAcceptors* got tokens that involve with hydrogen bonding, *NumAromaticRings* got the components of aromatic rings, and so on. We included this table in our Supplementary results.

property	top 15 SMILES tokens with highest attention scores, starting from the highest
NumHDonors	Br, (F)(F), C[NH+], CC[NH+], C(F)(F), (F)(F)F, CCCCCCCC, [N+](=O)[O-], I, NC(=O)N, c(F), C(=O)N, [NH+], (C(=O)N, C(F), ...
NumHAcceptors	[N+](=O)[O-], C(=O)N1, N2, NC(=O)N, CCCC, N1, [N+](=O)[O-], [nH], c(C(=O)N, (CC(=O)N, CCN(C(=O), CCCCCCCC, c1cccc1), S(=O)(=O)N, C(=O)N, ...
ExactMolWt	CCCCCCCC, (F)(F)F, C(F)(F), C(=O)N1, [n, S(=O)(=O), CCN(C(=O), (C(=O)N, CCCCC, I, Br), #, S(=O)(=O)N, O=C(N, N2, ...
RingCount	#, nn, #N), c2c1, =, [N, c(=O), S(=O), (N), (C(=O)N, =N, c2ccc3, [n, c3cccc3), c1ccc2c(c1), ...
NumAromaticRings	nn, n, cn, nc(, nc1, cn1, c3cccc3, COC(=O), ncc, c2c1, n1, Cn1, c2cccc2, =N,)cc2), ...
NumRotatableBonds	CCN(C(=O), CCCCCCCC, C(=O)N1, N1, Cc1ccc(, NC(=O)N, N2, S(=O)(=O)N, c1cccc1), c3cccc3, c1ccc2c(, c1ccc(C, CCCCC, CCC1, CCCCC1, ...
TPSA	Br, I, C(=O)[O-], =S, [NH+], Br, CCCCCCCC, C(F)(F), [n, (F)(F)F, CCCCC, C(=O)N1,)N1, [N+](=O)[O-], [N+](=O)[O-], ...
HeavyAtomCount	Br, I, CCCCCCCC, Br), C(F)(F), (F)(F)F, C(=O)N1, CCCCC, CCN(C(=O), [N+](=O)[O-]), S(=O)(=O), [S, c1cccc1),)N1, c2cccc2), ...
MolLogP	[S, (N), CCCCCCCC, s1, (F)(F)F), c(F), C(F)(F), F)cc1, Cn1, c3cccc, NC(=O)N, Br, c1cccc1), CCCCC, I, ...
MolMR	I, [N+](=O)[O-], Br), CCCCCCCC, Br, (F)(F)F, CCCCC, [N+](=O)[O-], C(F)(F), CCN(C(=O), CC(C)(C), [S, NC(=O)N, c(F), (C)C), ...
NOCount	e, N), I, [N+](=O)[O-], [N+](=O)[O-], O), C(=O)N, CCCCCCCC, c1cccc1), NC(=O)N, (N), 2)cc1, =[NH+], (F)(F), NC(=O), ...
QED	cccc, Cc1cccc1, (Cl), 2)cc1,)cc2), c1ccc2c(c1), Br, c1cccc, Br), c2cccc2), cs, (F)(F)F), c1cccc1,)ccc1, c1cccc1), ...

Figure 1. Top 15 SMILES tokens that showed highest cross-attention score for each property. Only 12 out of 53 properties are listed.

For the final comments on the performance, we pre-trained SPMM with more training data of 50 million molecules, which improved the overall performance to perform best on **5 out of 9** tasks in the case of the MoleculeNet downstream task. See the updated Table 3 below.

Dataset #data #task	regression[RMSE, ↓]					classification[AUROC in %, ↑]				SOTA ratio
	Delaney ESOL 1128 1	LIPO 4200 1	Freesolv 642 1	BACE 1513 1	Clearance 837 1	BBBP 2039 1	BACE 1513 1	Clintox 1478 2	SIDER 1427 27	
D-MPNN ⁴⁹	1.050±0.008	0.683±0.016	2.082±0.082	2.253*	49.754*	71.0±0.3	80.9±0.6	90.6±0.6	57.0±0.7	0/9
N-GramRF ¹⁹	1.074±0.107	0.812±0.028	2.688±0.085	1.318*	52.077*	69.7±0.6	77.9±1.5	77.5±4.0	66.8±0.7	0/9
N-GramXGB ¹⁹	1.083±0.082	2.072±0.030	5.061±0.744	-	-	69.1±0.8	79.1±1.3	87.5±2.7	65.5±0.7	0/9
PretrainGNN ⁵⁰	1.100±0.006	0.739±0.003	2.764±0.002	-	-	68.7±1.3	84.5±0.7	72.6±1.5	62.7±0.8	0/9
GROVER _{large} ²⁰	0.895±0.017	0.823±0.010	2.272±0.051	-	-	69.5±0.1	81.0±1.4	76.2±3.7	65.4±0.1	0/9
ChemRL-GEM ²¹	0.798±0.029	0.660±0.008	1.877±0.094	-	-	72.4±0.4	85.6±1.1	90.1±1.3	67.2±0.4	3/7
ChemBERTa-2 _(MTR-77M) ²¹	0.889*	0.798*	-	1.363*	48.515*	72.8*	79.9*	56.3*	-	0/8
MolFormer ⁵²	0.880±0.028	0.700±0.012	2.342±0.052	1.047±0.029	43.175±1.537	73.6±0.8	86.3±0.6	91.2±1.4	65.5±0.2	1/9
SPMM(w/o pre-train)	1.272±0.015	1.009±0.021	3.018±0.179	1.675±0.010	53.544±0.312	66.6±0.3	78.7±2.6	76.3±1.5	57.1±1.6	
SPMM	0.817±0.010	0.681±0.004	1.868±0.041	1.041±0.022	42.607±0.675	75.1±0.9	84.4±0.4	92.7±0.7	66.9±0.9	5/9

Table 3: Benchmark results on MoleculeNet downstream tasks. The best performance for each task was written in bold, and the second-best performance was underlined. For each task, we fine-tuned our model in four random seeds and recorded the mean and the standard deviation of those results. The benchmark model results were taken from ChemRL-GEM and ChemBERTa-2. *The standard deviation cannot be found in the source of the benchmark results. †Unofficial results, obtained from the official checkpoint under our data preparation.

For forward/retro reaction prediction tasks, we believe that top-1 accuracy is a more important metric than other top-k(k>1) accuracies since the accurate prediction of the reaction is important. Also, top-1 accuracy and top-k accuracy do not increase concurrently: We observed that top-k accuracy started to decrease after some point of training while top-1 accuracy is still increasing. In this respect, achieving the best top-1 accuracy in the forward reaction task and second-best top-1 accuracy in the retro-reaction prediction task is a notable performance.

Finally, regarding the overall performances of SPMM on single-modality downstream tasks, in contrast to our model that perform various tasks using a single foundational model, all of these competitive results with state-of-the-art single-modality pre-trained models are obtained from a dedicated single model which requires additional models when the number of the downstream tasks increases. The use of single foundation model for various downstream tasks is one of the most important properties of the recent success of foundational models, which is also confirmed for chemical domain by our SPMM model.

REVIEWER COMMENTS

Reviewer #1 (Remarks to the Author):

I have read the revised manuscript and the replies to my previous comments and those of the other reviewer. Thank you to the authors for careful consideration of the comments and taking the time to address them. The relevant comparisons are now in place to demonstrate when and why a practitioner might want to employ the SPMM method.

I understand the reviewer 2's concern regarding the order of PV. It's also appreciated that the authors conducted an ablation study in which they pre-trained the model by replacing the next property prediction in the pre-train objective with a masked property prediction similar to BERT, which can eliminate the predetermined order issue that the reviewer 2 was concerned about. To further evaluate the effect of PV order, I recommend the authors change the order of PV randomly and conduct three parallel experiments at least across all tasks. If the SPMM exhibits sustained and satisfactory performance similar to its previous outcomes, it suggests that the order of PV might not have a discernible effect. Those experimental results also can strongly support their claims.

NCOMMS-23-06428B

Reply to the Reviewers

General Comments

We thank the editor and the reviewers for the constructive reviews. Your comments have helped us a lot in improving the quality of manuscript. We would first like to briefly list the major changes that we have made to the manuscript.

1. We've conducted an additional analysis do demonstrate that the specific property order in the PV doesn't influence the overall performance of SPMM.

With the general modifications mentioned above, we have conducted additional experiments and made proper revisions to the manuscript. For point-to-point response, please see below.

Reply to the Reviewer1

Comments to author: I have read the revised manuscript and the replies to my previous comments and those of the other reviewer. Thank you to the authors for careful consideration of the comments and taking the time to address them. The relevant comparisons are now in place to demonstrate when and why a practitioner might want to employ the SPMM method.

We would like to thank the reviewer for thoughtful and constructive feedback.

R1.C1 I understand the reviewer 2's concern regarding the order of PV. It's also appreciated that the authors conducted an ablation study in which they pre-trained the model by replacing the next property prediction in the pre-train objective with a masked property prediction similar to BERT, which can eliminate the predetermined order issue that the reviewer 2 was concerned about.

To further evaluate the effect of PV order, I recommend the authors change the order of PV randomly and conduct three parallel experiments at least across all tasks. If the SPMM exhibits sustained and satisfactory performance similar to its previous outcomes, it suggests that the order of PV might not have a discernible effect. Those experimental results also can strongly support their claims.

Thanks for the constructive comments. Following the suggestion from the reviewer, we pre-trained two more separated SPMM models with different randomly-permuted property orders in the PV to demonstrate that the pre-defined order of the properties doesn't affect the performance of SPMM.

Specifically, we compared three SPMM models' performance for all tasks that we've done in the manuscript, and the overall performance was maintained regardless of the permutation of the property orders. Please refer to Table 1 below. We included this analysis in the manuscript and the Supplementary results.

PV-to-SMILES generation task											
model	sampling	input PV	Validity[↑]	Uniqueness[↑]	Novelty[↑]	normalized RMSE[↓]					
Permutation #1	deterministic	1,000 unseen PubChem SMILES' PV	0.995±0.001	0.999±0.001	0.961±0.005	0.216±0.004					
	stochastic	full PV of the molecule I Molecular weight=150	0.974±0.005	0.905±0.007	0.998±0.003	0.185±0.004					
Permutation #2	deterministic	1,000 unseen PubChem SMILES' PV	0.992±0.002	1.000±0.000	0.991±0.001	0.207±0.003					
	stochastic	full PV of the molecule I Molecular weight=150	0.941±0.005	0.909±0.011	0.997±0.01	0.170±0.002					
Permutation #3	deterministic	1,000 unseen PubChem SMILES' PV	0.996±0.001	1.000±0.000	0.990±0.002	0.193±0.003					
	stochastic	full PV of the molecule I Molecular weight=150	0.960±0.004	0.908±0.006	0.998±0.001	0.167±0.001					
SMILES-to-PV generation task			normalized RMSE[↓]	r^2 score[↑]							
			Permutation #1	0.128	0.923						
			Permutation #2	0.098	0.939						
			Permutation #3	0.110	0.925						
MoleculeNet	regression[RMSE, ↓]					classification[AUROC in %, ↑]					
Dataset	Delaney ESOL	LIPO	Freesolv	BACE	Clearance	BBBP	BACE	Clintox	SIDER		
Permutation #1	0.817±0.010	0.681±0.004	1.868±0.041	1.041±0.022	42.607±0.675	75.1±0.9	84.4±0.4	92.7±0.7	66.9±0.9		
Permutation #2	0.834±0.011	0.672±0.008	1.951±0.013	1.042±0.005	42.633±0.438	75.5±0.3	84.4±0.7	92.0±0.5	67.1±0.4		
Permutation #3	0.845±0.003	0.681±0.006	1.875±0.017	1.037±0.007	43.754±0.292	75.1±0.2	85.8±0.5	90.7±0.1	67.0±0.8		
DILI classification task		Acc in %[↑]	Selectivity in %[↑]	Specificity in %[↑]	AUROC in %[↑]						
		Permutation #1	84.4	83.9	84.6	92.6					
		Permutation #2	81.5	79.6	87.7	91.1					
		Permutation #3	84.2	83.5	80.0	91.6					
Reaction prediction tasks		forward acc.[↑]				retro acc.[↑]					
		top-1	top-2	top-3	top-5	top-1	top-5	top-10			
		Permutation #1	0.915	0.935	0.946	0.954	0.534	0.676	0.703		
		Permutation #2	0.913	0.934	0.945	0.952	0.538	0.668	0.699		
		Permutation #3	0.915	0.934	0.946	0.955	0.525	0.680	0.710		

Table 1. The performance comparison of three separately pre-trained SPMs that utilized their PV with different property orders. Permutation #1 is a property order of PV that was used for the main results.

REVIEWERS' COMMENTS

Reviewer #1 (Remarks to the Author):

Based on the performance comparison of three separately pre-trained SPMs that utilized their PV with different property order, the authors proved a fact that the order of PV not have a significant effect on their model. Completing these parallel experiments is not an easy task that requires a lot of time. Thank you to the authors for careful consideration of my comment and taking the time to address them. The manuscript is suggested to be published.

Thank you for considering my opinions.
Best wishes